# DP-LLM: Runtime Model Adaptation with Dynamic Layer-wise Precision Assignment

**Sangwoo Kwon, Seong Hoon Seo, Jae W. Lee**[*]**, Yeonhong Park**[*]
Seoul National University
{kwonsw055, andyseo247, jaewlee, ilil96}@snu.ac.kr
https://github.com/SNU-ARC/DP-LLM

## Abstract

How can we effectively handle queries for on-device large language models (LLMs) with varying runtime constraints, such as latency and accuracy? Multi-scale quantization addresses this challenge by enabling memory-efficient runtime model adaptation of LLMs through the overlaying of multiple model variants quantized to different bitwidths. Meanwhile, an important question still remains open-ended: how can models be properly configured to match a target precision or latency? While mixed-precision offers a promising solution, we take this further by leveraging the key observation that the sensitivity of each layer dynamically changes across decoding steps. Building on this insight, we introduce DP-LLM, a novel mechanism that dynamically assigns precision to each layer based on input values. Experimental results across multiple models and benchmarks demonstrate that DP-LLM achieves a superior performance-latency trade-off, outperforming prior approaches.

## 1 Introduction

Runtime model adaptation, which involves selecting models with different trade-offs between size (or inference latency) and accuracy, has been an important research topic for handling queries with varying runtime requirements, especially for on-device local LLM inference. Recently, multi-scale quantization techniques [1, 2] have been proposed as an effective solution for runtime adaptation of large language models (LLMs). While existing approaches for non-LLM DNNs typically maintain multiple separate models [3, 4, 5, 6, 7], this strategy is infeasible for LLMs due to memory constraints. Instead, multi-scale quantization enables the deployment of multiple LLM variants quantized to different bitwidths in a memory-efficient manner by effectively overlaying them.

Although multi-scale quantization provides an efficient framework for runtime model adaptation of LLMs, an important question remains open-ended: how to configure a model to effectively match a target precision or latency. For example, Any-Precision LLM [1], a multi-scale quantization framework, adapts by simply assigning a uniform precision across all layers. However, such a naive approach prevents support for models with non-integer precisions (e.g., 3.5-bit) and misses the opportunity to enhance model efficiency by mixing multiple bitwidths across layers.

While layer-wise mixed-precision [2, 8, 9] can be integrated with multi-scale quantization to address this issue, we identify an overlooked opportunity for further improvement: the sensitivity of each layer dynamically changes across decoding steps. In other words, a layer that requires a higher bitwidth at certain decoding steps due to its sensitivity may become less sensitive at other steps, making a lower bitwidth more suitable. This observation highlights the need for dynamic layer-wise precision assignment at each decoding step, rather than static precision assignment.

---

[*]Corresponding authors

39th Conference on Neural Information Processing Systems (NeurIPS 2025).

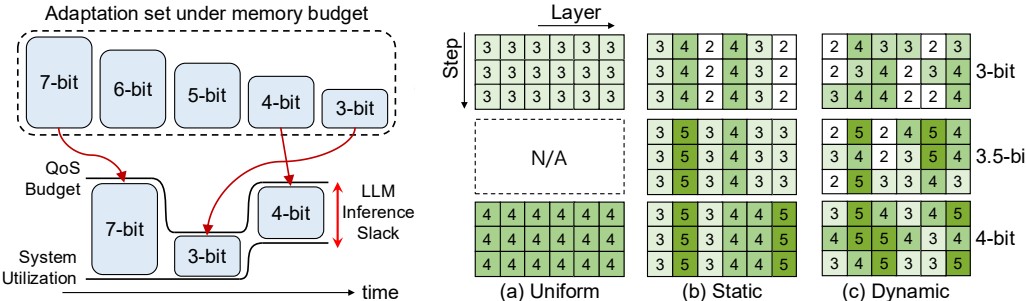

Figure 1: Runtime model adaptation

Figure 2: Different precision assignment schemes

To leverage this yet unexplored opportunity, we propose Dynamic-Precision LLM (DP-LLM), a model runtime adaptation mechanism that supports dynamic layer-wise precision assignment. DP-LLM defines a proxy metric of quantization sensitivity, called relative error, and determines for each layer, at offline, when to use high- or low-bit precision based on the magnitude of this proxy metric. At runtime, DP-LLM efficiently estimates the relative error and selects the appropriate precision for each layer. Through extensive experiments on various datasets and downstream tasks, we demonstrate that DP-LLM achieves a superior performance-latency trade-off compared to prior methods.

Our contributions are as follows:

- We find that layer-wise sensitivity to quantization is not a static property, but rather changes dynamically with each decoding step (token by token).
- We define a proxy metric, termed relative error, which serves as a criterion for precision assignment, and design a mechanism that learns, for each layer, how to determine which precision to apply based on the value of this metric.
- We develop a lightweight precision selector that efficiently estimates the relative error and selects the appropriate precision at runtime.
- We demonstrate that DP-LLM achieves superior performance on various datasets and downstream tasks under various memory and latency constraints, compared to prior works.

## 2 Background and Motivation

### 2.1 Quantization for On-Device LLM

LLM quantization approaches fall into two categories: weight-only quantization [10, 11, 12, 13] and weight-activation co-quantization [14, 15, 16, 17]. Since the low-batch characteristic of on-device LLM inference makes the weight memory access the main bottleneck [12, 18], the former is better suited for accelerating inference speed than the latter. OPTQ [10], AWQ [12], and SqueezeLLM [13] are some of the recently proposed schemes that fall into this category. However, none of the methods aforementioned can adapt to different query-wise runtime requirements, since the statically assigned precision cannot be changed at runtime.

### 2.2 Runtime Model Adaptation for LLMs

Different queries in DNN serving often have varying runtime constraints, such as accuracy, latency, or energy requirements. Edge devices are no exception, as the volatility in available resources [19, 20] and the variety of downstream tasks [21, 22] dynamically affect the runtime constraints. An effective approach to address these varying constraints is to maintain a set of models with different trade-offs between accuracy and overhead (e.g., latency, throughput, and energy), which we refer to as *adaptation set*, and select the most suitable one based on the runtime requirements for each query [3, 4, 5, 6, 7].

Figure 1 illustrates an example of runtime model adaptation. The adaptation set is configured under a given memory budget, and consists of models with different precisions. The gap between the varying query-wise resource budget under QoS (quality-of-service) constraints, or *QoS budget* for brevity, and the fluctuating system utilization creates volatility in the LLM inference slack, and at any given

time, the model with the precision that best fills the slack is selected. For example, when the QoS budget is relaxed and the system utilization is low, a higher-precision model such as a 7-bit variant is preferred. Conversely, when the QoS budget is tight and/or the system utilization is high—leaving limited slack—a lower-precision model (e.g., 3-bit or 4-bit) may be selected.

Such an approach, however, is inapplicable for on-device LLM inference scenarios, due to the insufficient memory capacity [23] to maintain multiple versions of the model and the high cost of training an LLM [24]. To address this issue, multi-scale quantization techniques [1, 2] have been proposed. This approach efficiently stores LLMs quantized to varying bitwidths (e.g., 3, 4, ..., $n$-bit) in a way that fits within the memory budget of a single $n$-bit model. Such techniques define the adaptation set as different bitwidth versions of a single LLM model, whose storage requirements are minimized by overlaying them efficiently.

## 2.3 Open Question: How to Configure an Adaptation Set

While multi-scale quantization provides a practical framework for implementing runtime model adaptation in LLM inference, it leaves open the question of how to configure an adaptation set. A naive approach of constructing an adaptation set with uniform integer precision models results in two notable sub-optimalities. First, this approach overlooks the well-known fact that not all model parameters are equally sensitive to quantization. Some parameters are more sensitive, whose approximation may lead to greater degradation in model quality than the approximation of others. Second, this approach restricts the adaptation set to coarse-grained configurations. In other words, it does not allow for options that fall between two integer bitwidth models, thereby limiting fine-grained control over latency-accuracy trade-offs. Such a fine-grained control is even more crucial at lower bits, where performance degradation is substantial.

**Solution: Layer-wise Mixed-Precision Adaptation.**  The aforementioned limitations of the uniform precision assignment strategy call for the adoption of mixed-precision schemes for configuring an adaptation set. Specifically, layer-wise mixed-precision schemes present a promising solution with minimal overhead. For example, layer-wise mixed precision schemes can seamlessly integrate with Any-Precision LLM [1], an efficient multi-scale quantization framework, whereas more fine-grained mixed-precision schemes (e.g., element-wise, channel-wise) would require significant modifications to the software engine, potentially leading to latency degradation.

In fact, various methods have previously been proposed for layer-wise mixed-precision in vision models [9, 25, 26], and this concept has recently been extended to LLMs [2, 8]. These methods typically involve performing sensitivity profiling for each layer via offline analysis using a calibration dataset. Based on the profiling results, higher bitwidths are assigned to layers deemed more sensitive, while lower bitwidths are assigned to less sensitive layers. By incorporating such methods, multi-scale quantization frameworks can support configurations with *effective* bitwidths corresponding to non-integer values while also potentially improving the quality of configurations corresponding to integer bitwidth models. Figure 2(b) visualizes how the incorporation of layer-wise mixed-precision can achieve these benefits compared to the uniform bitwidth strategy, as shown in Figure 2(a).

## 2.4 Opportunities for Dynamic Layer-wise Mixed-Precision

One aspect that has not received much attention regarding layer-wise mixed-precision in the context of LLM inference is the dynamically changing sensitivity of each layer during the decoding phase. That is, layers that are sensitive at certain decoding steps may become less sensitive at others, and vice versa. However, prior works assign the bitwidth of layers statically: once a specific bitwidth is assigned to a layer, it remains fixed throughout the decoding phase, failing to account for this dynamic behavior [2, 8, 9, 25, 26].

Figure 3(a) demonstrates such dynamically changing sensitivity of layers. This figure shows the distribution of the top 20% most sensitive layers at each decoding step when decoding with a Llama-3-8B model, using the first sample from the C4 dataset with only the first half layers depicted. Assuming a scenario where 3-bit and 4-bit precision are used to configure a mixed-precision model, the sensitivity of each layer is defined as the decrease in perplexity when applying 4-bit precision to that layer while keeping the rest at 3-bit. The distribution is highly irregular, which calls for dynamic

bitwidth assignment for each layer at each decoding step. Figure 2(c) depicts our proposed dynamic layer-wise mixed-precision.

Figure 3(b) shows the perplexity of a dynamic layer-wise mixed-precision model that dynamically chooses between 3-bit and 4-bit for each layer based on sensitivity at each decoding step, using the same sensitivity analysis method as in Figure 3(a). While the exact implementation of this scheme is practically infeasible—since figuring out sensitivity at runtime is not possible—it serves as an indicator of the potential improvements achievable through the concept of dynamic layer-wise mixed-precision. This scheme is compared to a static layer-wise mixed-precision scheme, which assigns bitwidth to layers statically based on their average sensitivity. Dynamic layer-wise mixed-precision demonstrates substantial improvements over the

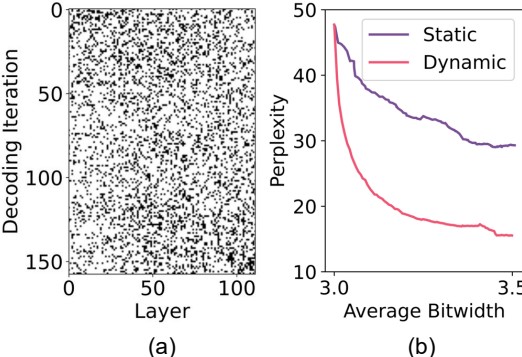

(a)          (b)

Figure 3: (a) Sensitivity of different layers at each decoding step. (b) Perplexity trend of different precision assignment schemes.

static approach, highlighting the importance of leveraging the dynamic behavior of layer-wise sensitivity.

## 3 DP-LLM Overview

Leveraging the dynamic nature of layer-wise sensitivity discussed in Section 2.4, we introduce Dynamic-Precision LLM (DP-LLM), a novel approach that employs a dynamic, layer-wise mixed-precision scheme for runtime model adaptation in LLMs. Figure 4 presents an overview of DP-LLM. Instead of preconfiguring the adaptation set offline, DP-LLM dynamically configures the model according to the target precision at runtime. Specifically, at offline, DP-LLM assigns each layer a *candidate precision set*, a set of precision levels that can be selected at runtime based on input values in each decoding step. To limit complexity, each candidate set consists of two precision levels, $h$-bit and $l$-bit, where $h > l$. To enable such runtime selection, DP-LLM augments each linear operation (essentially a GEMV operation when decoding with a batch size of 1) in the Transformer block with a *precision selector*. This precision selector processes the input vector and, based on its values, determines the appropriate precision.

The main insight behind designing the precision selector is that the norm of the difference in the GEMV output between using $W_h$ ($h$-bit weight) and $W_l$ ($l$-bit weight), or $||\Delta W x||$ (where $\Delta W = W_h - W_l$ and $x$ is the input vector), can serve as an effective indicator determining which precision to use. We refer to this difference in magnitude as *relative error*. Intuitively, it is preferable to use $W_l$ for inputs that result in small relative errors, while $W_h$ should be prioritized for inputs that are likely to produce larger relative errors.

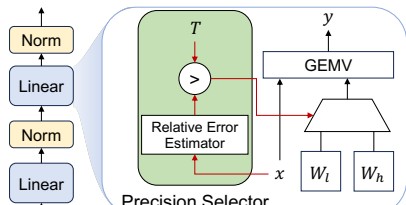

Figure 4: Overview of DP-LLM

Such an approach necessitates addressing two key challenges. The first challenge is determining a threshold $T$, along with $h$ and $l$ for each layer: inputs whose relative error exceeds $T$ use the higher-precision weights $W_h$, while those below $T$ use the lower-precision weights $W_l$. These $T$ values must be carefully determined, considering the impact of each layer on the end-to-end loss. Additionally, the $T$ values should be configured in a way that ensures the average bitwidth across the model closely matches the target precision. The second challenge lies in devising an efficient scheme to approximate the relative error with minimal computational overhead. While relative error guides our bitwidth selection, its exact computation is infeasible at runtime, as it would require performing an additional GEMV operation.

Section 4 explains how DP-LLM addresses the first challenge, while Section 5 explains how DP-LLM tackles the second.

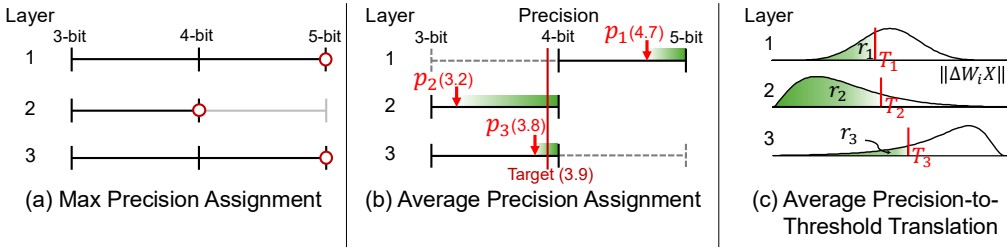

Figure 5: Overview of layer-wise candidate precision set and threshold assignment

# 4   Layer-wise Candidate Precision Set and Threshold Assignment

The process of determining candidate precision set and threshold (i.e., $h$, $l$ and $T$) for each layer is divided into three phases: (1) Layer-wise maximum precision selection (2) Layer-wise average precision assignment and (3) Average precision-to-threshold translation. Figure 5 provides an overview of these phases, while Algorithm 1 provides the overall procedures.

---

**Algorithm 1** Layer-wise Candidate Precision Set and Threshold Assignment of DP-LLM

---

**Input:** Calibration Dataset $D$, Multi-scale Quantized Model $\mathcal{M}$, Target Precision $b_{targ}$,
    Memory Budget $M_{max}$, Minimum Precision $b_{min}$
**Output:** Threshold list $T$

// Phase 1. Fit to memory budget
$S \leftarrow$ Calculate static sensitivity for each layer $i$
$B \leftarrow$ Find maximum precision for each layer $i$ using $S$, $M_{max}$

// Phase 2. Fine-tune average precision
**for** layer $i$ **in** $\mathcal{M}$ **do**
    Initialize $p_i$ for fine-tuning where $b_{min} \le p_i \le B[i]$
    Substitute linear operation $y = W_i x$ with $y = \sum_{b=b_{min}}^{B[i]} (s_{i,b} W_{i,b} x + t_{i,b} W_{i,b+1} x)$
    where $s_{i,b} = \begin{cases} 1 - (p_i - b) & (b \le p_i < b+1) \\ 0 & (otherwise) \end{cases}$ and $t_{i,b} = \begin{cases} p_i - b & (b < p_i \le b+1) \\ 0 & (otherwise) \end{cases}$
Fine-tune $\{p_i\}$ for $\mathcal{M}$ with $D$ using Equation 1 as loss function

// Phase 3. Translate average precision to threshold
**for** layer $i$ **in** $\mathcal{M}$ **do**
    $\Delta W_i \leftarrow W_{i,h} - W_{i,l}$ where $l = \lfloor p_i \rfloor$ and $h = \lceil p_i \rceil$
    Initialize err_list as empty list
    err_list.append($\|\Delta W_i x\|$) using $D$
    $r_i \leftarrow 1 - (p_i - l)$
    $T[i] \leftarrow r_i$-quantile of err_list

---

**Phase 1: Layer-wise Maximum Precision Selection.** DP-LLM adapts to the given memory budget by first selecting the maximum precision for each layer (List $B$ in Algorithm 1) before determining the average precisions. For example, in Figure 5(a), the maximum precisions for Layers 1, 2, and 3 are set to 5, 4, and 5, respectively. Inspired by [8, 9, 13], we use the second-order Taylor expansion on the loss function as the static sensitivity metric and form an integer programming problem to select maximum precisions. The detailed formulation is provided in Appendix A.

**Phase 2: Layer-wise Average Precision Assignment.** For dynamic precision selection, DP-LLM determines the average precision, $p$, for each layer. The average precision represents the expectation of selected precisions throughout the decoding phase for each layer. Once $p$ is determined, the corresponding candidate precision set is defined as $l = \lfloor p \rfloor$ and $h = \lceil p \rceil$. For example, a layer with a $p$ value of 3.2 would have a candidate precision set of 3-bit and 4-bit, and is expected to select 3-bit

weights for 80% of the decoding steps and 4-bit weights for 20% of them throughout the decoding phase.

DP-LLM assigns a distinct $p$ value to each layer considering the sensitivity of each layer. For example, in Figure 5(b), the target precision of 3.9 does not necessarily imply that $p = 3.9$ for all layers. Instead, different layers can have varying $p$ values (and varying $l$ and $h$ accordingly), as long as the overall average results in the target precision.

DP-LLM parameterizes $p$ values and performs fine-tuning to optimize these values with respect to the end-to-end loss. Specifically, during the fine-tuning process, the original linear layer $y = Wx$ is substituted with $y = rW_lx + (1 - r)W_hx$, where $l = \lfloor p \rfloor$, $h = \lceil p \rceil$, and $r = 1 - (p - l)$. This formulation, however, naturally causes the $p$ values to collapse to $h$ (and eventually to the highest precision possible), as using high bitwidth weights is generally more advantageous in terms of minimizing the loss. This outcome is undesirable, as it deviates from the target precision. To address this, we introduce a regularization term into the loss function to ensure that the overall average of $p$ values across all layers aligns with the target precision. Equation 1 shows the modified loss function, where $M_i$ is the number of parameters in layer $i$, $b_{targ}$ is the target precision, and $\alpha$ is a hyperparameter controlling the regularization strength.

$$\mathcal{L}' = \mathcal{L} + \alpha \left( \sum_i^N p_i M_i \bigg/ \sum_i^N M_i \ - \ b_{targ} \right)^2 \tag{1}$$

This fine-tuning process is conducted on a small calibration dataset with only a few iterations. In addition, the $p$ values are the only parameters updated during the fine-tuning process. Thus, both memory and computational costs are kept at a manageable level. An analysis of the fine-tuning cost is provided in Appendix B.3.

**Phase 3: Average Precision-to-Threshold Translation.** Once the average precisions have been assigned to each layer, they are translated into threshold values through statistical analysis using the calibration dataset—the same data samples used for average precision assignment. Specifically, DP-LLM leverages the distribution of relative errors from the calibration dataset to approximate the distribution of runtime inputs. For each layer, $||\Delta W_i x||$ is calculated during the calibration stage. Then, among the distribution of relative errors, the $r_i$-quantile value is selected as the threshold, where $r_i = 1 - (p_i - l)$. Figure 5(c) visualizes this process.

## 5 Relative Error Estimation

### 5.1 Hybrid Approach for Relative Error Estimation

DP-LLM adopts a hybrid approach for relative error estimation, where each layer selects one of two methods: linear regression-based or random projection-based estimation. The former is highly lightweight but relies on a strong linear relationship between the input vector norm $||x||$ and the relative error $||\Delta Wx||$ for accurate estimation. Hence, it is applied only to layers exhibiting such a relationship. For the remaining layers, the latter method is used; although less lightweight, it performs robustly across diverse scenarios. To determine the strength of the linear relationship, the coefficient of determination ($R^2$) between the input vector norm and the relative error for each layer is computed using the calibration set, and compared against a hyperparameter $R_{th}^2$ (which is set to 0.9 in our further experiment setups) at offline.

**Linear Regression-Based Estimation** For layers where $R^2$ exceeds $R_{th}^2$, and thus is considered to have a strong linear relationship, the relative error is approximated as a simple linear function of the input vector norm. Formally, $||\Delta Wx|| \approx ||x|| \times \alpha + \beta$, where $\alpha$ and $\beta$ are found by fitting a linear model using the calibration set. This approach incurs near-zero latency and GPU memory overhead. Notably, approximately half of the layers satisfy this property.

**Random Projection-Based Estimation** For layers without a strong correlation, DP-LLM utilizes random projection leveraging the Johnson-Lindenstrauss Lemma [27] (JL Lemma) to efficiently perform relative error estimation in a low-dimensional space. JL Lemma states that the probability of satisfying Inequality 2 for any $n$-dimensional vector $x$ is $1 - \delta$ when $A$ is a $k \times n$ matrix sampled

from a normal distribution $A_{ij} \sim \frac{1}{\sqrt{k}} N(0, 1)$ and $k = \mathcal{O}(\epsilon^{-2} \log(\delta^{-1}))$. In other words, estimating a vector's norm with a degree of confidence can be done by calculating the norm of the multiplication of a randomized matrix and the vector. In our case of estimating the input token's error, we need to estimate the norm of $\Delta W x$. Therefore, we plug in $\Delta W x$ to the lemma to get Inequality 3.

$$(1 - \epsilon)||x||_2^2 \leq ||Ax||_2^2 \leq (1 + \epsilon)||x||_2^2 \tag{2}$$

$$(1 - \epsilon)||\Delta W x||_2^2 \leq ||A\Delta W x||_2^2 \leq (1 + \epsilon)||\Delta W x||_2^2 \tag{3}$$

By precomputing $G = A\Delta W$, the relative error can be estimated at runtime by performing a small GEMV operation on $G$ and $x$, which is an $\mathcal{O}(nk)$ operation. In our further experiment setups, we use $k = 64$ for every linear projection, which limits the relative error estimation difference within 15% with 91% confidence when measured empirically using the C4 dataset. The error difference can be further reduced by calibrating $G$ with the distribution of $x$. Using the calibration set, we tune $G$ to match the input distribution at offline. This procedure can be performed in parallel for each layer, and only takes a small fraction of time (<10 seconds per layer).

**GPU Memory Overhead.** The additional GPU memory overhead in DP-LLM primarily stems from storing $G$ matrices for layers that use the random projection-based estimation. For these layers, DP-LLM must maintain a separate $G$ matrix for each pair of $l$ and $h$. For instance, consider a scenario where multiple target precisions ranging from 3-bit to 6-bit are supported. This results in three $(l, h)$ pairs: $(3, 4)$, $(4, 5)$ and $(5, 6)$. In the case of Phi-3-Medium, the sum of the $G$ matrices for each individual pair is approximately 0.17GB. Thus, for three pairs, the total GPU memory overhead amounts to around 0.51GB, corresponding to a 4.3% increase in GPU memory usage (assuming the model in 6-bit precision requires about 12GB). Since this represents the worst-case scenario where all layers rely on the random projection-based estimators, the space overheads in real-world scenarios are much smaller, which are reported in Appendix B.3.

## 5.2 Asynchronous Estimation

Although lightweight, it is undesirable that DP-LLM's relative error estimation, which requires the immediate input of each layer, lies on the critical path of inference. To minimize slowdown, as depicted in Figure 6, DP-LLM performs asynchronous estimation by using the previous input vector instead of the immediate one whenever possible. Recent studies [28, 29] have demonstrated that activations in Transformer architectures change slowly across blocks due to residual connections. This creates an

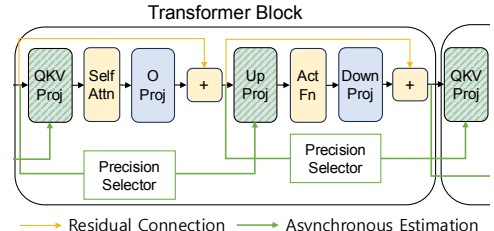

Figure 6: Asynchronous estimation

opportunity for layers directly connected to residual connections to use the previous residual output as the input for error estimation. Specifically, query, key, value, and up-projection correspond to such layers. The latency of asynchronous estimation can be masked by overlapping it with other layer computations.

## 6 Experiments

We implement DP-LLM on top of Any-Precision LLM [1], a multi-scale weight-only quantization framework for runtime model adaptation of LLMs. This section presents the performance benefits of DP-LLM (Section 6.1) and evaluates its latency overheads (Section 6.2). Lastly, we analyze the impact of DP-LLM's dynamic layer-wise precision selection on per-query QoS (Section 6.3).

### 6.1 Performance Evaluation

**Methodology.** We evaluate DP-LLM's performance on two open-source LLMs: Llama-3-8B [30] and Phi-3-Medium (14B) [31]. We evaluate perplexity on Wikitext2 [32] and C4 [33] datasets. Additionally, we evaluate decoding performances using datasets with sufficient token generation lengths. Specifically, we utilize GSM8K [34], MBPP [35], BBH [36], and MATH [37]. DP-LLM is

Table 1: Perplexity evaluation results

| Dataset | | WikiText2 (↓) | | | | | | | C4 (↓) | | | | | | |
|---|---|---|---|---|---|---|---|---|---|---|---|---|---|---|---|
| | | Target Precision (Bits) | | | | | | | Target Precision (Bits) | | | | | | |
| Models | Method | 3.25 | 3.50 | 3.75 | 4.00 | 4.25 | 4.50 | 4.75 | 3.25 | 3.50 | 3.75 | 4.00 | 4.25 | 4.50 | 4.75 |
| L3-8B | LLM-MQ | 7.62 | 7.38 | 7.21 | 7.07 | 6.94 | 6.83 | 6.73 | 12.01 | 11.66 | 11.42 | 11.21 | 10.97 | 10.78 | 10.66 |
| | HAWQ-V2 | 7.47 | 7.21 | 7.01 | 6.83 | 6.67 | 6.56 | 6.44 | 11.77 | 11.41 | 11.09 | 10.76 | 10.45 | 10.28 | 10.06 |
| | DP-LLM | **7.35** | **7.00** | **6.77** | **6.59** | **6.49** | **6.41** | **6.37** | **11.57** | **11.03** | **10.65** | **10.25** | **10.10** | **9.97** | **9.89** |
| P3-M | LLM-MQ | 5.26 | 5.17 | 5.09 | 5.04 | 4.87 | 4.74 | 4.56 | 9.57 | 9.49 | 9.45 | 9.40 | 9.34 | 9.22 | 9.09 |
| | HAWQ-V2 | 5.18 | 5.06 | 4.94 | 4.83 | 4.72 | 4.57 | 4.47 | 9.45 | 9.38 | 9.28 | 9.21 | 9.13 | 9.05 | 9.00 |
| | DP-LLM | **5.02** | **4.81** | **4.70** | **4.61** | **4.55** | **4.49** | **4.43** | **9.30** | **9.17** | **9.10** | **9.03** | **9.00** | **8.97** | **8.95** |

fine-tuned to find the layer-wise threshold by using 1000 samples from the C4 train dataset. Each sample is tokenized and the first 512 tokens of each sample are used. For DP-LLM, the hybrid approach for error estimation is applied for all layers, while asynchronous estimation is only applied to applicable layers. Detailed setups for benchmarks are available at Appendix B.1.

To demonstrate the effectiveness of DP-LLM's dynamic layer-wise precision selection, for a given memory budget and target precision pair, we compare DP-LLM against static layer-wise mixed-precision policies, which are also built on top of Any-Precision LLM for comparison. For static mixed-precision baselines, we adopt two methods: LLM-MQ [8] and HAWQ-V2 [9]. LLM-MQ utilizes the gradients of weights to estimate the loss perturbation ($\Delta\mathcal{L} \approx g^T \Delta W$), while HAWQ-V2 uses the second-order information ($\Delta\mathcal{L} \approx \overline{Tr}(H)\|\Delta W\|_2^2$). The detailed configurations for the two methods are provided in Appendix B.2. For all methods, we assume the Any-Precision LLM model that supports from 3-bit to 6-bit, selecting 6-bit as the upper limit since higher precision yields only marginal performance improvements on most datasets. All methods initially select precisions layer-wise to match the memory budget, then create adaptation sets to match the target precision utilizing the selected configuration. For the prefill phase in downstream task evaluations, we use the highest available precision for each layer, as lower precision does not provide benefits in this phase.

**Results.** We show the performance of each precision assignment scheme under various memory and target precision configurations. Specifically, we evaluate models by incrementally increasing the target precision in 0.25-bit steps within the available precision range. Tables 1 and 2 present perplexity and downstream task evaluation results for target precisions between 3-bit and 5-bit under 5-bit memory budgets, where mixed-precision schemes show meaningful differences. Evaluation results under other memory budgets are available in Appendix C.

Across most datasets and model pairs, DP-LLM demonstrates a superior performance compared to the static precision allocation schemes in various configurations. This demonstrates the effectiveness of dynamic precision selection, and validates our approach of capturing the sensitivity dynamics at runtime by utilizing relative error as an indicator.

Meanwhile, the static baselines fail to capture the dynamic nature of sensitivity, and are only able to utilize static sensitivity information given at the calibration phase—gradient of weights for LLM-MQ and Hessian matrix for HAWQ-V2. Thus, the static baselines cannot adapt to the varying sensitivity at runtime and show suboptimal performance.

Table 3: Perplexity measurement of exact error estimator

| Target Precision | | 3.5 | 4.0 | 4.5 |
|---|---|---|---|---|
| Wiki | Exact | 6.98 | 6.56 | 6.40 |
| | Approx. | 7.00 | 6.59 | 6.41 |
| C4 | Exact | 11.00 | 10.23 | 9.97 |
| | Approx. | 11.03 | 10.25 | 9.97 |

**Impact of Approximation.** To identify the effect of the relative error estimator, we calculate the perplexity with the estimator replaced with an exact error estimator. Although impractical, the exact error estimator calculates $\|\Delta W x\|$ without any approximation or asynchronous estimation, serving as an upper bound. Table 3 shows the measurement results using Llama-3-8B. Our approximation techniques show comparable performance, while minimizing computational costs.

Table 2: Downstream task evaluation results

| Dataset | | GSM8K (↑) | | | | | | | MBPP (↑) | | | | | | |
|---|---|---|---|---|---|---|---|---|---|---|---|---|---|---|---|
| | | Target Precision (Bits) | | | | | | | Target Precision (Bits) | | | | | | |
| Models | Method | 3.25 | 3.50 | 3.75 | 4.00 | 4.25 | 4.50 | 4.75 | 3.25 | 3.50 | 3.75 | 4.00 | 4.25 | 4.50 | 4.75 |
| L3-8B | LLM-MQ | 33.1 | 35.6 | 38.4 | 38.8 | 40.4 | 41.2 | 41.3 | 45.0 | 45.4 | 48.2 | 48.9 | 48.9 | 48.5 | 46.6 |
| | HAWQ-V2 | **37.7** | 38.2 | 41.0 | **43.7** | **45.8** | 44.2 | 45.2 | 47.5 | 47.8 | **50.8** | 48.9 | 51.1 | 52.2 | 50.8 |
| | DP-LLM | 36.7 | **39.4** | **42.2** | 42.8 | 45.6 | **45.7** | **46.9** | **47.8** | 50.4 | 49.4 | **49.2** | **51.3** | **52.9** | **53.6** |
| P3-M | LLM-MQ | 80.2 | 81.2 | 81.3 | 82.3 | 82.2 | 82.6 | 82.3 | 65.8 | **66.3** | 62.5 | 62.3 | 61.1 | 62.5 | 63.5 |
| | HAWQ-V2 | 81.0 | 79.8 | 81.0 | 81.9 | 82.4 | 83.2 | 83.1 | **67.9** | 66.3 | 62.5 | 63.9 | 65.3 | 65.3 | 62.8 |
| | DP-LLM | **81.6** | **82.0** | **83.9** | **84.3** | **83.9** | **84.2** | **83.3** | 66.0 | 65.8 | **68.6** | **67.2** | **69.1** | **66.5** | **64.9** |

| Dataset | | BBH (↑) | | | | | | | MATH (↑) | | | | | | |
|---|---|---|---|---|---|---|---|---|---|---|---|---|---|---|---|
| L3-8B | LLM-MQ | 43.9 | 45.6 | 46.9 | 47.5 | 47.9 | 48.9 | 48.3 | **11.2** | 9.0 | 9.4 | 11.2 | 11.6 | 11.8 | 11.8 |
| | HAWQ-V2 | 44.5 | 45.4 | 47.8 | **49.1** | **50.0** | 48.9 | 50.0 | 10.0 | 10.2 | 10.0 | **13.4** | 13.4 | 14.0 | 14.4 |
| | DP-LLM | **46.3** | **47.0** | **48.4** | 48.5 | 49.0 | **50.2** | **50.6** | 10.6 | **12.0** | **12.8** | 12.8 | **15.0** | **15.4** | **14.8** |
| P3-M | LLM-MQ | **57.3** | 57.1 | 57.6 | 57.7 | 57.8 | **58.5** | 58.4 | 30.4 | 29.6 | 30.4 | 31.0 | 31.0 | 31.6 | **34.2** |
| | HAWQ-V2 | **57.3** | 57.1 | 57.7 | 57.8 | 58.0 | **58.5** | 58.4 | 30.8 | 31.4 | 29.4 | 31.4 | 32.6 | 33.0 | **34.2** |
| | DP-LLM | **57.3** | **57.9** | **58.1** | **58.5** | **58.3** | 58.3 | **58.9** | **32.0** | **33.0** | **32.4** | **32.2** | **33.6** | **34.6** | **34.2** |

## 6.2 Inference Latency Evaluation

**Methodology.** We measure the inference latency overhead introduced by DP-LLM's precision selector. Specifically, we implement DP-LLM on top of gpt-fast [38], which optimizes it using the `torch.compile` feature [39]. To isolate the overhead of the precision selector and exclude the impact of varying effective bitwidth, we fix the effective bitwidth and compare the latency against a static precision allocation baseline (LLM-MQ) without the precision selector. The evaluation is conducted on two hardware platforms: NVIDIA Jetson Orin AGX 64GB [40] and NVIDIA RTX 4060 Ti 16GB [41].

**Latency Evaluation.** Table 4 shows the latency overheads, while Table 5 shows the average Time-Per-Output-Token (TPOT). With an overall geomean of 1.45% for Llama-3-8B and 0.81% for Phi-3-Medium, the relative error estimator of DP-LLM incurs minimal latency increase. Furthermore, DP-LLM shows latency improvements proportional to the precision decrements. This suggests that DP-LLM can accurately translate the target precision into model inference with the expected latency within a small error margin.

Table 4: Overhead of runtime estimation normalized to the latency of static method

| Effective Bitwidth | | 3.25 | 3.50 | 3.75 | 4.00 | 4.25 | 4.50 | 4.75 | Geomean |
|---|---|---|---|---|---|---|---|---|---|
| L3-8B | Jetson | 2.39% | 2.30% | 3.48% | 3.62% | 2.83% | 3.46% | 4.24% | 3.12% |
| | 4060Ti | 1.56% | 0.74% | 0.66% | 0.66% | 0.58% | 0.45% | 0.53% | 0.68% |
| P3-M | Jetson | 2.33% | 1.83% | 2.71% | 4.10% | 2.58% | 0.99% | 0.06% | 1.32% |
| | 4060Ti | 1.67% | 0.84% | 0.08% | 1.14% | 1.12% | 0.44% | 0.13% | 0.50% |

Table 5: TPOT of DP-LLM

| Effective Bitwidth | | 3.25 | 3.50 | 3.75 | 4.00 | 4.25 | 4.50 | 4.75 | FP16 |
|---|---|---|---|---|---|---|---|---|---|
| L3-8B | Jetson | 28.77ms | 30.18ms | 31.87ms | 33.49ms | 34.67ms | 36.23ms | 37.81ms | 86.36ms |
| | 4060Ti | 15.54ms | 16.26ms | 17.09ms | 17.94ms | 18.78ms | 19.62ms | 20.47ms | 55.43ms |
| P3-M | Jetson | 43.84ms | 46.49ms | 49.73ms | 53.48ms | 56.37ms | 58.79ms | 61.05ms | 158.73ms |
| | 4060Ti | 23.33ms | 24.83ms | 26.31ms | 28.18ms | 28.89ms | 31.28ms | 32.95ms | OOM |

**Latency Ablation Study.** Table 6 demonstrates how the hybrid use of a linear regression-based estimation, instead of solely relying on a random projection-based estimation, along with asynchronous estimation, reduces the latency overhead of the precision selector. The ablation study is conducted using Llama-3-8B. In all cases, both techniques substantially improve inference speed.

Table 6: Latency overhead of relative error estimation techniques

|  | Jetson Orin AGX | | | RTX 4060Ti | | |
|---|---|---|---|---|---|---|
| Effective Bitwidth | 3.5 | 4.0 | 4.5 | 3.5 | 4.0 | 4.5 |
| Random Projection Based | 5.04% | 5.87% | 5.74% | 3.94% | 3.53% | 3.09% |
| Hybrid | 3.36% | 4.23% | 4.37% | 1.61% | 1.39% | 1.09% |
| Hybrid+Async | 2.30% | 3.62% | 3.46% | 0.74% | 0.66% | 0.45% |

## 6.3 Per-Query QoS Validation

Since DP-LLM aims to match the overall average precision to the target precision on a best-effort basis, individual queries may exhibit deviations from the target precision. To assess the

Table 7: Per-query effective bitwidth increase

| Target Precision | 3.5 | 4.0 | 4.5 |
|---|---|---|---|
| 90th Percentile | 1.42% | 0.90% | 1.28% |
| 99th Percentile | 3.02% | 2.25% | 3.32% |

impact of DP-LLM on query-level QoS, we collect the distribution of effective bitwidth when inferencing queries in the Alpaca dataset [42], which consists of handwritten instructions for chatbot. The results, summarized in Table 7, report the increase of the 90th and 99th percentile effective bitwidth relative to the mean effective bitwidth. Even at the 99th percentile, the geomean increase remains below 3%, demonstrating that DP-LLM has minimal adverse impact on per-query QoS.

## 7 Conclusion

We propose DP-LLM, a runtime model adaptation scheme that enables dynamic layer-wise precision assignment. Based on our observation that the layer-wise sensitivity to precision varies across decoding steps, DP-LLM devises a novel mechanism that assigns precision dynamically at each decoding step based on a per-layer relative error threshold and a lightweight precision selector. DP-LLM achieves state-of-the-art performance across a wide range of datasets and configurations by leveraging the unrecognized opportunity of dynamism in sensitivity.

## Acknowledgments

This work was supported by Samsung Electronics Co., Ltd. and also by the following Institute of Information & Communications Technology Planning & Evaluation (IITP) grants: Artificial Intelligence Innovation Hub (No. 2021-0-02068) and Global Scholars Invitation Program (RS-2024-00456287), both funded by the Korea government (MSIT).

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

## A  Formulation of Layer-wise Maximum Precision Selection

To select the precision of each layer under a memory budget constraint, we follow the approach inspired by [8], [9], and [13], using a second-order Taylor expansion of the loss function to formulate an integer programming problem. Equation 4 presents the second-order Taylor expansion of the model's loss function, where $g$ and $H$ denote the gradient and the Hessian for $W$, respectively.

$$\mathcal{L}(W_Q) \approx \mathcal{L}(W) - g^T(W - W_Q) + \frac{1}{2}(W - W_Q)^T H(W - W_Q) \tag{4}$$

Assuming the model has converged, the first-order term can be ignored, allowing the second-order term to approximate the loss difference caused by quantization. Furthermore, by neglecting cross-weight interactions, only the diagonal elements of the Hessian ($H$) remain relevant. As a result, the loss difference can be expressed as shown in Equation 5.

$$\mathcal{L}(W_Q) - \mathcal{L}(W) \approx \frac{1}{2}(W - W_Q)^T diag(H)(W - W_Q) = \frac{1}{2}\sum H_{k,k}((W - W_Q)_k)^2 \tag{5}$$

When the weights of each layer $W_i$ are quantized to a $b$-bit weight $W_{i,b}$, the objective function to minimize can be expressed as Equation 6. Here, $N$ denotes the number of layers, $B$ is the set of available precisions, $M_i$ is the memory usage of the $i$th layer, and $b_{targ}$ is the target precision.

$$\underset{c_i,b}{\arg\min} \sum_i^N \sum_b^B c_{i,b} \cdot \sum_k H_{k,k}((W - W_{i,b})_k)^2 \tag{6}$$

$$s.t.\ \ c_{i,b} \in \{0,1\},\ \sum_b^B c_{i,b} = 1,\ \sum_i^N \sum_b^B c_{i,b} \cdot b \cdot M_i \le b_{targ} \cdot \sum_i^N M_i$$

To keep computational costs reasonable, we approximate $H$ using the Fisher information matrix $F$, whose diagonal elements can be calculated by accumulating the squared gradients over the calibration set. After solving Equation 6 via integer programming, $c_{i,b}$ indicates whether $b$-bit weight was selected for layer $i$. We then set the maximum precision for each layer accordingly.

## B  Evaluation Details

### B.1  Setup and Methodology

**Perplexity Datasets.**  For both the WikiText2 and C4 datasets, samples are concatenated and divided into chunks of size 2048. The perplexity evaluation follows the same procedure as Any-Precision LLM [1]. We interpret the perplexity evaluation as a teacher-forced decoding process, computing the loss at each decoding step to determine the overall perplexity.

**Downstream Tasks.**  GSM8K is evaluated in a 5-shot setting using the LM-evaluation-harness framework [43], with `exact_match` as the evaluation metric. MBPP is evaluated in a 3-shot setting using instruction-embedded prompts. The reported metric for MBPP is `pass@1`. The user instruction and assistant response prefix strings are adopted from [44]. An example MBPP prompt is shown below, with `[BEGIN]` and `[DONE]` tokens used to delimit the model's response.

```
1   You are an expert Python programmer, and here is your task:
2       Write a function to find the shared elements from the given two lists.
3
4   Your code should pass these tests:
5       assert set(similar_elements((3, 4, 5, 6),(5, 7, 4, 10))) == set((4, 5))
6       assert set(similar_elements((1, 2, 3, 4),(5, 4, 3, 7))) == set((3, 4))
7       assert set(similar_elements((11, 12, 14, 13),(17, 15, 14, 13))) == set((13,
            14))
8   [BEGIN]
```

Figure 7: Example MPPP Prompt

BBH is evaluated in a 3-shot setting using the LM-evaluation-harness framework, with Chain-of-Thought (CoT) reasoning enabled. The reported metric is `exact_match`. MATH is evaluated using the MATH-500 [45] variant, also with 3-shot prompting. The evaluation metric for MATH is `math_verify`.

**Fine-tuning.** A subset of the C4 train dataset (512 tokens×1000 samples) is used for threshold fine-tuning, with the number of epochs and learning rate each set to 5 and 0.01, respectively. Fine-tuning is performed using the AdamW optimizer. The hyperparameter $\alpha$ is set to 1 for all target precisions, except when the target precision is 3.25, where $\alpha$ is set to 10 to better align with the target precision. The fine-tuning framework is built on top of the Any-Precision LLM [1] codebase, enabling the use of quantized weights for efficient fine-tuning.

**Latency Measurements.** To ensure full GPU saturation during the decoding phase, by building on top of gpt-fast [38], each model is compiled using PyTorch's `torch.compile` function. This compiles a CUDA Graph of the model, eliminating the kernel launch overheads and improving GPU utilization efficiency. Latency is measured by generating 100 tokens across 10 repetitions, with the average effective bitwidth set equal to the target precision. This bitwidth is fixed by setting each layer's threshold to either infinity or negative infinity.

## B.2 Static Precision Assignment Baseline.

LLM-MQ formulates the precision assignment problem as follows:

$$\underset{c_{i,b}}{\operatorname{argmin}} \sum_i^N \sum_b^B c_{i,b} \cdot |g_i^T (W_i - W_{i,b})| \tag{7}$$

$$s.t. \ c_{i,b} \in \{0,1\}, \ \sum_b^B c_{i,b} = 1, \ \sum_i^N \sum_b^B c_{i,b} \cdot b \cdot M_i \leq b_{targ} \cdot \sum_i^N M_i$$

where $N$ denotes the number of layers, $B$ is the set of available precisions, $W_i$ is the original weight of layer $i$, $W_{i,b}$ is the $b$-bit quantized weight of layer $i$, $g_i$ is the gradient of $W_i$, $M_i$ is the memory usage of layer $i$, and $b_{targ}$ is the target precision. However, since the original constraint only enforces an upper bound on memory usage, the scheme often fails to find a feasible allocation for higher precision targets (e.g., 4.5 bits) due to the similarity of high precision weights. To address this, we introduce a lower bound into the constraint formulation:

$$\sum_i^N \sum_b^B c_{i,b} \cdot b \cdot M_i \geq b_{targmin} \cdot \sum_i^N M_i \tag{8}$$

$b_{targmin}$ is gradually increased from zero to $b_{targ}$ in increments of 0.01, until the average memory usage falls within 0.005 bits of the target precision.

HAWQ-V2 uses the second-order information to calculate the sensitivity of a layer:

$$\Omega_i = \overline{Tr(H)} ||W - W_Q||_2^2 \tag{9}$$

Since computing the exact Hessian matrix is impractical for an LLM, we approximate it using the Fisher information matrix, following the approach in [13]. Once the sensitivities are computed, we formulate an integer programming problem to identify the optimal adaptation set, as done in LLM-MQ.

## B.3 Fine-tuning Results

**Fine-tuning Cost.** Both Llama-3-8B and Phi-3-Medium are fine-tuned on a single RTX 3090 GPU with 24GB of VRAM. Llama-3-8B completes fine-tuning in approximately one hour, utilizing 14GB of VRAM, while Phi-3-Medium takes about two hours with 21GB of VRAM usage. On an A100 80GB GPU, Llama-3-8B requires about 30 minutes, and Phi-3-Medium takes approximately one hour to complete fine-tuning.

**Distribution of Average Precision.** We present examples of fine-tuned average precisions for target precisions of 3.5 and 4.0 bits under a 5-bit memory budget. Figures 8 and 9 show the results for Llama-3-8B, while Figure 10 and 11 correspond to Phi-3-Medium. These results demonstrate that the average precisions are distributed across the full range of available values, rather than being concentrated at the lower or upper extremes.

**Relative Error Estimation Method Selection.** Table 8 presents the number of layers assigned to each error estimation method for every $h$ and $l$ pair. For Llama-3-8B, nearly half of the layers use linear regression for error estimation, which introduces negligible overhead and significantly reduces GPU memory and latency overhead compared to relying solely on random projections based on the Johnson-Lindenstrauss (JL) Lemma (see Section 6.2). In the case of Phi-3-Medium, approximately two-thirds of the layers are estimated using linear regression.

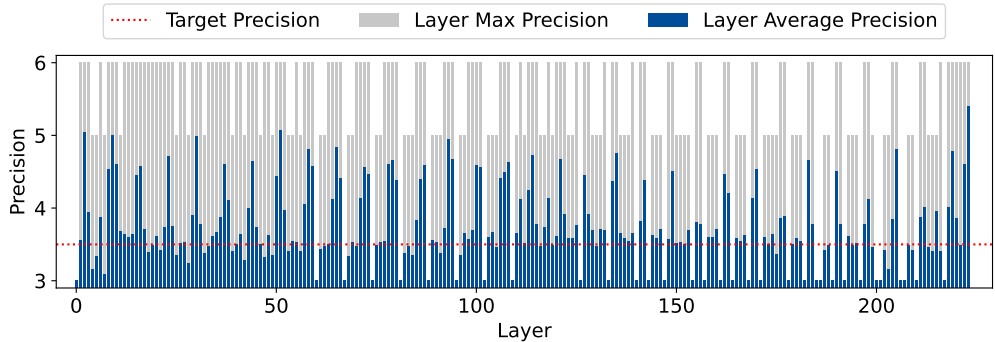

Figure 8: Average precisions for Llama-3-8B with target precision of 3.5 bits

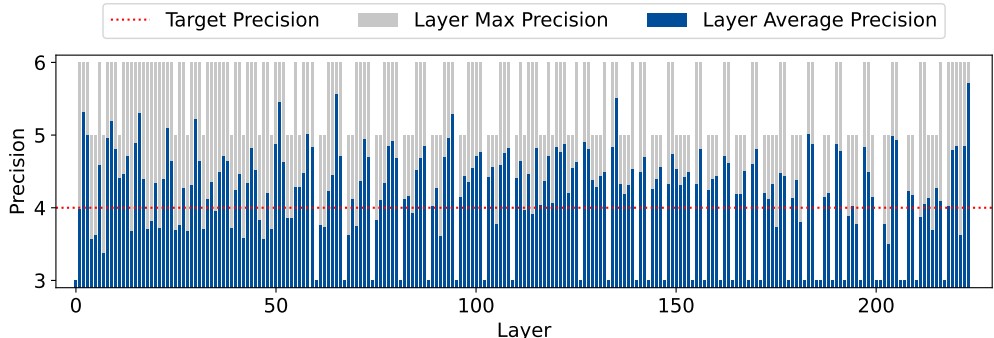

Figure 9: Average precisions for Llama-3-8B with target precision of 4.0 bits

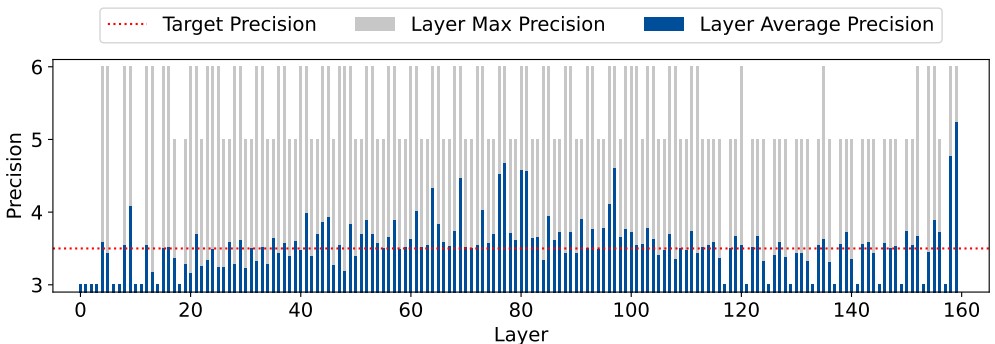

Figure 10: Average precisions for Phi-3-Medium with target precision of 3.5 bits

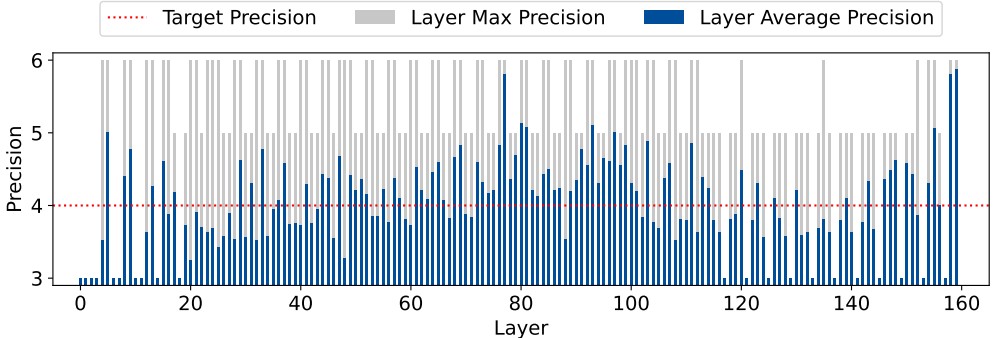

Figure 11: Average precisions for Phi-3-Medium with target precision of 4.0 bits

Table 8: Number of layers for each relative error estimation method

| $l, h$ | 3, 4 | | 4, 5 | | 5, 6 | |
|---|---|---|---|---|---|---|
| Model | Linear | JL | Linear | JL | Linear | JL |
| Llama-3-8B | 107 | 117 | 113 | 111 | 111 | 113 |
| Phi-3-Medium | 115 | 45 | 110 | 50 | 108 | 52 |

**GPU Memory Overhead**   The combined GPU memory overhead is measured by summing the additional capacity required for relative error estimators that support runtime model adaptation across multiple target precisions (3.25, 3.5, 3.75, 4.0, 4.25, 4.5, 4.75). The results are presented in Table 9. The overhead is computed relative to a baseline model constrained by a 5-bit memory capacity budget, consistent with the main evaluation.

Table 9: GPU memory overhead of DP-LLM

| Model | Llama-3-8B | Phi-3-Medium |
|---|---|---|
| Quantized Model Capacity | 6.7GB | 9.4GB |
| Average Estimator Capacity Per Target Precision | 0.08GB | 0.07GB |
| Total Estimator Capacity | 0.16GB | 0.12GB |
| Overhead | 2.4% | 1.3% |

## C   Additional Evaluations

### C.1   Evaluations on Different Memory Budgets

We evaluate DP-LLM under varying memory budgets, specifically at 6-bit and 4-bit configurations. Since the baseline Any-Precision model supports precisions ranging from 3 to 6 bits, all layers are permitted to utilize any available precision under the 6-bit memory budget. Table 10 presents the evaluation results for the 6-bit budget, while Table 11 shows the results for the 4-bit budget.

Table 10: Evaluation results under 6-bit memory budget

| Dataset | | WikiText2 ($\downarrow$) | | | | | C4 ($\downarrow$) | | | | |
|---|---|---|---|---|---|---|---|---|---|---|---|
| | | Target Precision (Bits) | | | | | Target Precision (Bits) | | | | |
| Models | Method | 3.5 | 4.0 | 4.5 | 5.0 | 5.5 | 3.5 | 4.0 | 4.5 | 5.0 | 5.5 |
| | LLM-MQ | 7.27 | 7.07 | 6.63 | 6.65 | 6.32 | 11.48 | 11.21 | 10.40 | 10.50 | 9.81 |
| L3-8B | HAWQ-V2 | 7.21 | 6.83 | 6.56 | 6.32 | 6.20 | 11.41 | 10.76 | 10.28 | 9.81 | **9.60** |
| | DP-LLM | **7.02** | **6.59** | **6.39** | **6.25** | **6.19** | **11.06** | **10.28** | **9.91** | **9.67** | 9.60 |
| | LLM-MQ | 5.14 | 4.85 | 4.77 | 4.53 | 4.39 | 9.47 | 9.26 | 9.22 | 9.07 | 8.97 |
| P3-M | HAWQ-V2 | 5.06 | 4.83 | 4.57 | **4.40** | **4.36** | 9.38 | 9.21 | 9.05 | 8.95 | 8.93 |
| | DP-LLM | **4.82** | **4.60** | **4.50** | 4.42 | **4.36** | **9.17** | **9.04** | **8.97** | **8.93** | **8.92** |

### C.2   Evaluations on Models with Different Parameter Scales

We further evaluate DP-LLM on the Qwen2.5-3B and Qwen2.5-32B models to examine how model size impacts its performance. Table 12 presents the evaluation results under a 5-bit memory budget. The results show that DP-LLM consistently achieves superior performance across different model sizes.

### C.3   Ablation study for selecting $h$ and $l$

DP-LLM selects $h = \lceil p \rceil$ and $l = \lfloor p \rfloor$ as the high and low precisions, respectively, to construct an average precision $p$. While multiple combinations of $h$ and $l$ are possible, we empirically find that choosing values close to the target precision yields the best model performance. Table 13 shows the fine-tuning results under various $h$ and $l$ combinations when targeting a 4.5-bit precision. To isolate the effects of these choices, all layers are constrained to use the same $h$ and $l$ pair during fine-tuning. The results clearly show that selecting neighboring precisions to the target value is most effective for constructing an average precision.

Table 11: Evaluation results under 4-bit memory budget

| Dataset | | WikiText2 (↓) | | | C4 (↓) | | |
|---|---|---|---|---|---|---|---|
| | | Target Precision (Bits) | | | Target Precision (Bits) | | |
| Models | Method | 3.25 | 3.50 | 3.75 | 3.25 | 3.50 | 3.75 |
| | LLM-MQ | 7.62 | 7.39 | 7.21 | 12.01 | 11.66 | 11.42 |
| L3-8B | HAWQ-V2 | 7.47 | 7.21 | 7.01 | 11.77 | 11.41 | 11.09 |
| | DP-LLM | **7.38** | **7.08** | **6.91** | **11.64** | **11.19** | **10.92** |
| | LLM-MQ | 5.26 | 5.18 | 5.06 | 9.57 | 9.52 | 9.45 |
| P3-M | HAWQ-V2 | 5.18 | 5.06 | 4.94 | 9.45 | 9.38 | 9.28 |
| | DP-LLM | **5.06** | **4.95** | **4.88** | **9.34** | **9.26** | **9.21** |

Table 12: Evaluation results for models with different parameter scales

| Dataset | | WikiText2 (↓) | | | | | | | C4 (↓) | | | | | | |
|---|---|---|---|---|---|---|---|---|---|---|---|---|---|---|---|
| | | Target Precision (Bits) | | | | | | | Target Precision (Bits) | | | | | | |
| Models | Method | 3.25 | 3.50 | 3.75 | 4.00 | 4.25 | 4.50 | 4.75 | 3.25 | 3.50 | 3.75 | 4.00 | 4.25 | 4.50 | 4.75 |
| | LLM-MQ | 9.83 | 9.58 | 9.34 | 9.29 | 9.07 | 9.07 | 8.89 | 16.07 | 15.72 | 15.40 | 15.36 | 15.06 | 15.06 | 14.83 |
| Q2.5-3B | HAWQ-V2 | 9.34 | 9.00 | 8.77 | 8.59 | 8.42 | 8.30 | **8.21** | 15.50 | 15.01 | 14.67 | 14.40 | 14.20 | 14.02 | 13.92 |
| | DP-LLM | **9.13** | **8.83** | **8.58** | **8.43** | **8.33** | **8.27** | **8.21** | **15.17** | **14.73** | **14.42** | **14.25** | **14.05** | **13.95** | **13.88** |
| | LLM-MQ | 5.76 | 5.66 | 5.56 | 5.51 | 5.48 | 5.43 | 5.30 | 10.92 | 10.83 | 10.76 | 10.73 | 10.70 | 10.67 | 10.59 |
| Q2.5-32B | HAWQ-V2 | 5.68 | 5.51 | 5.40 | 5.30 | 5.23 | 5.17 | 5.13 | 10.85 | 10.71 | 10.62 | 10.55 | 10.50 | 10.46 | **10.43** |
| | DP-LLM | **5.57** | **5.40** | **5.28** | **5.21** | **5.17** | **5.12** | **5.10** | **10.79** | **10.64** | **10.54** | **10.51** | **10.48** | **10.45** | **10.43** |

Table 13: Perplexity under different $l$ and $h$ combinations

| Dataset | | WikiText2 (↓) | C4 (↓) |
|---|---|---|---|
| Model | $l$ & $h$ | Perplexity | Perplexity |
| | 3 & 5 | 6.55 | 10.30 |
| | 3 & 6 | 6.78 | 10.65 |
| Llama-3-8B | **4 & 5** | **6.36** | **9.87** |
| | 4 & 6 | 6.43 | 9.98 |
| | 3 & 5 | 4.49 | 9.03 |
| | 3 & 6 | 4.56 | 9.11 |
| Phi-3-Medium | **4 & 5** | **4.45** | **8.99** |
| | 4 & 6 | 4.48 | 9.01 |

## C.4 Calibration Set Sensitivity Study

We investigate the impact of the calibration set on model performance. Table 14 presents the evaluation results for Llama-3-8B when the calibration set is switched from the training set of C4 to that of WikiText2. The results indicate that in most cases, DP-LLM performs effective fine-tuning regardless of the choice of calibration set, without signs of overfitting.

Table 14: Evaluation results with different calibration sets

| Dataset | WikiText2 (↓) | | | | | | | C4 (↓) | | | | | | |
|---|---|---|---|---|---|---|---|---|---|---|---|---|---|---|
| | Target Precision (Bits) | | | | | | | Target Precision (Bits) | | | | | | |
| Calibration Set | 3.25 | 3.50 | 3.75 | 4.00 | 4.25 | 4.50 | 4.75 | 3.25 | 3.50 | 3.75 | 4.00 | 4.25 | 4.50 | 4.75 |
| WikiText2 | 7.32 | 6.98 | 6.75 | 6.60 | 6.50 | 6.41 | 6.36 | 11.59 | 11.05 | 10.68 | 10.33 | 10.16 | 10.00 | 9.91 |
| C4 | 7.35 | 7.00 | 6.77 | 6.59 | 6.49 | 6.41 | 6.37 | 11.57 | 11.03 | 10.65 | 10.25 | 10.10 | 9.97 | 9.89 |

# D   Limitations

DP-LLM is designed to enhance only the decoding phase of on-device LLM inference and does not affect the prefill phase. Additionally, downstream tasks evaluated solely based on log-probabilities are excluded from the experiments, as they do not involve decoding. DP-LLM aims to meet QoS constraints on a best-effort basis; QoS-guaranteed dynamic precision selection is left as future work.

# E   Decoding Examples

We present examples where DP-LLM successfully generates a response while the static baseline fails. DP-LLM dynamically assigns higher precision to decoding steps where the estimated output difference is significant, potentially preventing errors that could derail the entire decoding process. In the examples, the points of divergence are highlighted in red.

---

**Input Prompt (GSM8K)**

**Question:**
    Gretchen has 110 coins. There are 30 more gold coins than silver coins.
    How many gold coins does Gretchen have?

---

**Generated Responses**

**FP16 Original Model**

Let x be the number of silver coins Gretchen has. Then the number of gold coins is x+30.
The total number of coins is 110, so x+x+30=110 2x+30=110 2x=110-30 2x=80 x=80/2 x=40
So Gretchen has 40 silver coins and 40+30=«40+30=70»70 gold coins.
#### 70

**Static (HAWQ-V2)**

Let x be the number of gold coins. Then the number of silver coins is x+30.
The total number of coins is x+x+30=«x+x+30=2x+30»2x+30 So 2x+30=110, and 2x=80, so x=«2x=80, so x=40»40
#### 40

**DP-LLM**

Let x be the number of silver coins Gretchen has. Then the number of gold coins is x+30.
The total number of coins is 110, so x+x+30=110 2x+30=110 2x=80 x=40
So the number of gold coins is 40+30=«40+30=70»70
#### 70

---

**Input Prompt (BBH)**

**Question:**
The first day of 2019 is a Tuesday, and today is the first Monday of 2019.
What is the date 24 hours later in MM/DD/YYYY?

**Options:**
(A) 01/08/1960
(B) 01/01/2019
(C) 02/08/2018
(D) 01/08/2019
(E) 10/22/2018
(F) 01/08/2043

**Generated Responses**

**FP16 Original Model**

Let's think step by step.
The first day of 2019 is a Tuesday, and today is the first Monday of 2019. So today's date is Jan 7, 2019. The date 24 hours later is Jan 8, 2019, that is 01/08/2019. So the answer is (D).

**Static (HAWQ-V2)**

Let's think step by step.
The first day of 2019 is a Tuesday, and today is the first Monday of 2019. So today's date is Jan 1, 2019. 24 hours later is Jan 2, 2019. So the answer is (B).

**DP-LLM**

Let's think step by step.
The first day of 2019 is a Tuesday, and today is the first Monday of 2019. So today's date is Jan 7, 2019. 24 hours later is Jan 8, 2019, that is 01/08/2019. So the answer is (D).

