# OpenReview forum: "DP-LLM: Runtime Model Adaptation with Dynamic Layer-wise Precision Assignment"
_NeurIPS.cc/2025/Conference — NeurIPS 2025 poster_

### Official Review · Reviewer_sm1G · 2025-06-25

**Clarity:** 3
**Significance:** 2
**Originality:** 3
**Rating:** 4
**Confidence:** 5

**Summary:**

The paper proposes DP-LLM, a dynamic precision assignment framework for on-device LLM inference. Traditional multi-scale quantization techniques statically assign bitwidths across all or per-layer configurations. DP-LLM improves upon this by dynamically adjusting layer-wise precision at each decoding iteration based on runtime input sensitivity.

**Questions:**

1. Layer-wise maximum precision is determined in Appendix A. Does the B[i] equal to b where $c_{i,b}=1$? The constraint in (6) guarantees that when we use b where $c_{i,b}=1$, the memory usage for quantized model still below the memory budget. Why we need to further determine layer precision, but can't direct quantize each layer to its maximum precision? Is there some misunderstanding?

2. How to guarantee the fair comparison with baselines? Though G requires mild GPU memory overhead, we can also achieve performance improvement by protecting a very small fraction of weights in FP16. How does DP-LLM perform when have exactly storage cost with baselines.

3. The latency evaluation is not that straightforward. Methods like GPTQ, SqueezeLLM usually measure the seconds for generation of tokens and compared with FP16 models to show the efficiency. How does DP-LLM perform?

**Ethical Concerns:**

["NO or VERY MINOR ethics concerns only"]

**Final Justification:**

Thank you for the authors' response. As my original rating is already positive, I have decided to keep it unchanged.

**Limitations:**

See Cons and Questions.

**Quality:**

3

**Strengths And Weaknesses:**

Pros:

1. The paper identifies a previously overlooked behavior: the dynamic change in sensitivity of layers during token decoding.

2. DP-LLM Integrates dynamic bitwidth switching in a practical and low-overhead manner.

3. Experiment results shows significant improvements in perplexity and task accuracy over baselines, and ablation studies on latency and approximation show that design choices are well-justified.

Cons:

1. Lack of experiment on larger LLMs, the scalability of LLM quantization methods. Previous works like  AWQ, SqueezeLLM scale to 70B models. Given the complex architecture of DP-LLM, the complexity and cost in finetuning and inference in larger PLMs should be demonstrated.

2. Lack of low bit evaluation. The performance degradation becomes more obvious and a greater challenge in low-precision (as demonstrate in QuiP for 2-bit). There are also works like [2] show that quantize the whole layer into 2 bit will have a huge performance degradation. How DP-LLM performs in such scenario? Will the performance of DP-LLM highly depend on the Any-precision LLM method?

3. Lack of comparison on integer-bit. As a mixed-precision solution, it's also important to show the performance compared to existing works on integer-bit quantization task. [3] shows that SliM-LLM (which is also a mixed-precision method) achieve 6.4 ppl on wiki2 and 9.5 ppl on c4 on 4bit task for LLaMa3-8B. The comparison of such methods should be discussed.




[1] Chee, Jerry, et al. "Quip: 2-bit quantization of large language models with guarantees." Advances in Neural Information Processing Systems 36 (2023): 4396-4429.

[2] Chen, Zihan, et al. "Channel-wise mixed-precision quantization for large language models." arXiv preprint arXiv:2410.13056 (2024).

[3] Huang, Wei, et al. "An empirical study of llama3 quantization: From llms to mllms." Visual Intelligence 2.1 (2024): 36.

---

> ### Author Rebuttal · Authors · 2025-07-31
>
> Thank you for your expert evaluation and specific suggestions.
>
> ---
>
> > **#1**. The deployment of DP-LLM on larger LLMs needs evaluation.
>
> We evaluate the performance, fine-tuning costs, and inference overheads of DP-LLM when applied to Llama-3-70B. We measure the perplexity and compare it against HAWQ-V2 when the target precision is 3.5.
>
> |  | WikiText2 | C4 |
> |:---:|:---:|:---:|
> | HAWQ-V2 | 4.02 | 8.07 |
> | DP-LLM | **3.89** | **7.97** |
>
> On a single A100 80GB GPU, the fine-tuning of Llama-3-70B takes 3.5 hours. The cost of fine-tuning the average precision increases sublinearly with respect to the parameter size, considering that Llama-3-8B takes 0.5 hours to fine-tune on the same GPU (as reported in Appendix B.3).
>
> We evaluate the inference latency overhead on Jetson Orin AGX 64GB, the only edge device with large enough memory to load a 70B model. The latency overhead is kept minimal even when the model size increases. When the effective bitwidth is 3.5, the latency overhead could not be observed when measuring tokens per second up to the second decimal place, which was measured as 4.56 tok/s. This is possible because the complexity of the relative error estimator increases linearly with the hidden dimensions ($\mathcal O(nk)$, where $k=64$ for our error estimator setup), while the linear layers have approximately quadratic complexities since most Transformer architectures scale both the hidden dimension and the intermediate dimension.
>
> ---
>
> > **#2**. Does the performance of DP-LLM depend on Any-Precision LLM? Also, low-bit evaluation scenarios are lacking.
>
> DP-LLM is not inherently tied to Any-Precision LLM. Its core idea—dynamic, layer-wise precision assignment based on saliency estimation—is orthogonal to the underlying quantization method and can be used in conjunction with any backend that supports multiple bitwidths. We adopt Any-Precision LLM primarily for its memory efficiency, as it enables runtime model adaptation without requiring multiple copies of the model. However, if memory capacity is not a constraint, DP-LLM can also be used with other quantization approaches by separately maintaining multiple bitwidth variants in memory. Under tight memory constraints, Any-Precision LLM is a natural choice to support runtime adaptation; however, this reliance is not unique to DP-LLM and similarly applies to other approaches such as static mixed-precision methods.
>
> Decoupled from Any-Precision LLM, we evaluate DP-LLM with a state-of-the-art quantization method to demonstrate its effectiveness in the low-bit regime (e.g., 2.x-bit). The table below reports the perplexity of DP-LLM on LLaMA-3-8B, where it dynamically selects among 2- to 4-bit weights quantized using GuidedQuant [1] in comparison to static mixed-precision baselines. In all cases, DP-LLM consistently outperforms the static approaches, demonstrating its robustness under aggressive quantization.
>
> |  | WikiText2 |  |  | C4 |  |  |
> |:---:|:---:|:---:|:---:|:---:|:---:|:---:|
> | Target Precision | 2.25 | 2.5 | 2.75 | 2.25 | 2.5 | 2.75 |
> | LLM-MQ | 20.4 | 15.64 | 11.87 | 21.34 | 17.23 | 15.24 |
> | HAWQ-V2 | 16.85 | 11.61 | 9.40 | 18.80 | 15.03 | 13.30 |
> | DP-LLM | **14.06** | **10.87** | **8.92** | **17.02** | **14.55** | **12.89** |
>
> [1] Kim et al., “GuidedQuant: Large Language Model Quantization via Exploiting End Loss Guidance”, ICML, 2025.
>
> ---
>
> > **#3**. Comparison with SliM-LLM.
>
> We evaluate SliM-LLM when the effective bitwidth is 4, using the same perplexity measuring tools as DP-LLM to ensure consistency.
>
> |  | WikiText2 | C4 |
> |:---:|:---:|:---:|
> | SliM-LLM | **6.41** | 10.47 |
> | DP-LLM | 6.59 | **10.25** |
>
> While DP-LLM archives comparable performance against SliM-LLM, it is worth noting the following limitations of SliM-LLM.
>
> First, unlike DP-LLM, the precision reduction of SliM-LLM does not translate into speedup, which is crucial for runtime model adaptation. In particular, SliM-LLM reports to have 5.15% to 27.06% slowdown compared to GPTQ, often being slower than the FP16 baseline. DP-LLM, on the other hand, shows minimal latency overhead from 0.06% to 4.24% as shown in Table 4.
>
> Second, SliM-LLM assigns a uniform target precision for every layer and only performs mixed precision allocation to groups within each layer. On the other hand, DP-LLM can assign different target average precisions to each layer, providing more flexibility in precision assignment.
>
> ---
>
> > **#4**. Is $B[i]$ equal to $b$ where $c_{i,b}=1$? Also, the use of layerwise average precision is puzzling.
>
> Yes, $B[i]$ equals $b$ where $c_{i,b}=1$.
>
> The runtime model adaptation calls for the distinction between layer-wise maximum precision and layer-wise average precision.
>
> The layer-wise maximum precision is determined to fit the model into the memory capacity budget of the device. Since DP-LLM requires high precision weights to be available for dynamic precision selection, leaving high precision weights that are not important would be wasteful, or even cause out-of-memory situations under tight memory budgets.
>
> The layer-wise average precision, on the other hand, is used for fitting the model for the current runtime budget (i.e., LLM inference slack of Figure 1). After the model is loaded into memory, to perform runtime model adaptation, the layer-wise average precision is used to match the current runtime budget.
>
> ---
>
> > **#5**. Comparison with sparse outlier protection methods is needed to address fair comparison under equal storage costs.
>
> We apply the methodology of OWQ [2], a state-of-the-art quantization method that saves additional weight channels in FP16, to harvest outlier channels for each layer. We set the number of outliers to match the storage overhead of DP-LLM, which is 2.4% for Llama-3-8B as reported in Table 8. We then measure the perplexity after adding the outlier channels to the static baselines.
>
> |  | WikiText2 |  |  | C4 |  |  |
> |:---:|:---:|:---:|:---:|:---:|:---:|:---:|
> | Target Precision | 3.5 | 4.0 | 4.5 | 3.5 | 4.0 | 4.5 |
> | LLM-MQ (w/ Outliers) | 7.19 | 6.94 | 6.73 | 11.36 | 10.98 | 10.61 |
> | HAWQ-V2 (w/ Outliers)  | 7.06 | 6.74 | 6.51 | 11.21 | 10.64 | 10.20 |
> | DP-LLM | **7.00** | **6.59** | **6.41** | **11.03** | **10.25** | **9.97** |
>
> Although saving outliers shows notable performance improvements, DP-LLM still consistently outperforms the static baselines by a considerable margin.
>
> It is also worth noting that outlier protection comes at the cost of latency overhead. We measure the slowdown of OWQ incurred by the outliers under the same configuration. Since the GitHub repository of OWQ reports that the kernels were optimized for A100 GPUs, we evaluate the kernel time on an A100 GPU using PyTorch Profiler. We find that outlier protection incurs an average of 18.42% kernel time slowdown. Applying a finer-grained outlier protection (e.g., element-wise) would lead to even greater slowdown.
>
> [2] Lee et al., “OWQ: Outlier-Aware Weight Quantization for Efficient Fine-Tuning and Inference of Large Language Models”, AAAI, 2024.
>
> ---
>
> > **#6**. Clarification on latency evaluation.
>
> We present the time-per-output token (TPOT) values under several effective bitwidths. We also report the TPOT of the full precision (i.e., FP16) model for comparison.
>
>
> | Device | Effective Bitwidth | 3.5 | 4.0 | 4.5 | FP16 |
> |:---:|:---:|:---:|:---:|:---:|:---:|
> | RTX 4060Ti | Llama-3-8B | 16.26ms | 17.94ms | 19.62ms | 55.43ms |
> |  | Phi-3-Medium | 24.83ms | 28.18ms | 31.28ms | OOM |
> | Jetson | Llama-3-8B | 30.18ms | 33.49ms | 36.23ms | 86.36ms |
> |  | Phi-3-Medium | 45.64ms | 51.28ms | 58.21ms | 158.73ms |
>
> By employing low-overhead error estimation techniques, DP-LLM shows latency improvements proportional to the precision decrements.

---

> > ### Comment · Reviewer_sm1G · 2025-08-05
> >
> > Thank you for the authors' response. As my original rating is already positive, I have decided to keep it unchanged.

---

### Official Review · Reviewer_WtEe · 2025-07-01

**Clarity:** 2
**Significance:** 2
**Originality:** 3
**Rating:** 4
**Confidence:** 3

**Summary:**

The paper proposes a lightweight mechanism that extends AnyPrecisionLLM to enable dynamic layerwise precision selection based on their saliency metric. AnyPrecisionLLM (prior work) solves the problem of storing multi-precision weights while DP-LLM (introduced here) proposes a solution for dynamic, fine-grained bit width allocation.

**Questions:**

- It is unclear what precision activations are assumed to be at. Are those dynamically selected as well, or fixed at high precision?
- In lines 231-237, it is stated that the choice of $k$ in the JL projection can limit the "relative error estimation difference within 8% with 95% confidence". Is this an empirical or theoretical statement?
- In section 6.3, I didn't quite understand why the geometric mean was being reported. Could you please explain? My concern is that the geometric mean can be quite small relative to the arithmetic mean and can hide outliers.

**Ethical Concerns:**

["NO or VERY MINOR ethics concerns only"]

**Final Justification:**

I appreciated the effort by the authors to address the issues I raised. While I am not 100% convinced about the need for, hence impact of the work, I am comfortable with increasing my score to 4.

**Limitations:**

Please see questions and weaknesses.

**Paper Formatting Concerns:**

/

**Quality:**

3

**Strengths And Weaknesses:**

Strengths:

- While mixed-precision quantization is increasingly being studied, the problem of dynamic, fine-grained bit width allocation does not seem well-explored. The insight that layer importance may change over decoding steps is interesting.
- I thought the idea of approximating $||\Delta W x||_2$ by using the Johnson-Lindenstrauss lemma was quite nice.


Weaknesses:

- While the insight that layer importance may change over decoding steps is interesting, it is only briefly discussed and loosely analyzed. It would be great to have a deeper analysis and discussion.
- As a more general concern, the paper is not particularly well positioned in the wider context and history of quantization, and model optimization in general. For example, fine-grained saliency estimation has been studied for quite a while, but this is not addressed in the paper.
- The experiment design does not take into account the current state-of-the-art in LLM quantization.

To expand on some of these issues, saliency metrics are well studied, and have been since the 90's including in "Optimal Brain Surgeon" in 1993, "Deep Compression" in 2016, and LLM “super weights” in 2024. Additionally, the paper only benchmarks against 2 static mixed-precision techniques and does not consider cheaper techniques that are known to be sufficient saliency estimators (e.g., weight and/or activation magnitude). As such, it is unclear if the paper’s proposal (which introduces compute and memory overhead) can be undercut by a simpler metric.

While it is clear that query budgets can change, there is not enough discussion or analysis on how much the budget can vary. This is foundational of the utility of the work, so this should be part of the motivation.

Furthermore, techniques such as Once-for-All in 2019 and EagleEye in 2020 propose methods for encoding several mixed-precision models and architectures, similar to AnyPrecisionLLM, but more fine-grained. If the goal is to adjust precision to query budgets, this seems like a natural benchmark as well, since these techniques can instead adjust neural architecture based on inference cost constraints.

Finally, when benchmarking against static precision allocation, one should benchmark against the state-of-the-art techniques for PTQ (e.g., SpinQuant, OPTQ, GPFQ, OSTQuant, etc.) and QAT (ParetoQ). Although these papers study uniform precision, a primary contribution of this paper is saliency estimation, which should be compatible with static bit width allocation as well. Given that, for example, SpinQuant + OPTQ can yield high quality sub-4-bit quantized models in a PTQ setting and that ParetoQ models are trained to use 2-bit or even 1.58-bit weights, it seems reasonable that one can use these techniques to statically optimize a model to the tightest runtime constraints and circumvent the need for dynamic bit width allocation.

---

> ### Author Rebuttal · Authors · 2025-07-31
>
> Thank you for your constructive feedback.
>
> ---
>
> > **#1**. A quantitative and deeper analysis on dynamically changing sensitivity of layers (Section 2.4).
>
> We kindly ask to refer to our response of **#1** to Reviewer **QAUB**.
>
> ---
>
> > **#2**. The paper is not well positioned in the wider context and history of quantization.
>
> DP-LLM fundamentally differs from prior works on saliency metrics for model compression by addressing the _dynamic_ nature of saliency (referred to as _sensitivity_ in our manuscript). Previous approaches, such as Optimal Brain Surgeon [1], Deep Compression [2], and LLM Super Weights [3], primarily focus on static methods for estimating saliency at the layer or finer granularity. None of the existing methods capture dynamically changing sensitivity during inference, nor do they propose runtime techniques to handle such dynamics with minimal overhead. DP-LLM is, to the best of our knowledge, the first to address this issue. We will include this discussion on the positioning of DP-LLM within the broader context in the revised version.
>
> [1] Hassibi et al., “Second order derivatives for network pruning: Optimal Brain Surgeon”, NIPS, 1992.
>
> [2] Han et al., “Deep Compression: Compressing Deep Neural Networks with Pruning, Trained Quantization and Huffman Coding”, ICLR, 2016.
>
> [3] Yu et al., “The Super Weight in Large Language Models”, arXiv, 2024.
>
> ---
>
> > **#3**. Cheaper saliency estimators are not considered.
>
> We evaluate a simple saliency estimator based on activation norm [4, 5] and compare it to our proposed estimator based on relative error norm. Specifically, we evaluate perplexity on WikiText2 and C4 using Llama-3-8B. To configure the norm-based saliency estimators, for each dataset, we collect the distribution of the desired norm, and for each layer and decoding step, select the top-k% to apply a 4-bit weight instead of a 3-bit weight. We experimented with k=20% and k=50%, which roughly correspond to constructing a 3.2-bit and 3.5-bit model, respectively.
>
> | Ratio |  | WikiText2 | C4 |
> |:---:|:---:|:---:|:---:|
> |  | 3bit | 8.31 | 13.07 |
> | 20% | Activation norm | 8.01 | 12.62 |
> |  | Relative Error norm | **7.92** | **12.45** |
> | 50% | Activation norm | 7.57 | 11.90 |
> |  | Relative Error norm | **7.47** | **11.66** |
>
> It is clear that the relative error norm-based selection shows superior performance compared to the activation norm-based selection. This can be attributed to the fact that the relative error norm is a direct indicator of the quantization error, while the activation norm is an indirect measure of it.
>
> [4] Lin et al., “AWQ: Activation-aware Weight Quantization for On-Device LLM Compression and Acceleration”, MLSys, 2024.
>
> [5] Zhang et al., “LQER: Low-Rank Quantization Error Reconstruction for LLMs”, ICML, 2024.
>
> ---
>
> > **#4**. Only two static mixed-precision baselines were evaluated.
>
> As DP-LLM is designed for efficient runtime model adaptation with regard to fluctuating LLM inference slacks, the mixed precision scheme should have minimal latency overheads. More fine-grained mixed precision methods that employ channel-wise or element-wise mixed precision schemes cannot satisfy this constraint. For example, SliM-LLM [6] utilizes fine-grained block-wise mixed precision, but reports to have 5.15% to 27.06% slowdown compared to GPTQ, even often showing slower speeds than the FP16 baseline. The static baselines we evaluated, LLM-MQ and HAWQ-V2, employ a layer-wise mixed precision scheme, allowing for near-zero overhead, making them suitable baselines for evaluating the runtime adaptation ability of DP-LLM.
>
> [6] Huang et al., “SliM-LLM: Salience-Driven Mixed-Precision Quantization for Large Language Models”, ICML, 2025.
>
> ---
>
> > **#5**. Deploying a highly optimized model for the tightest margin could nullify the need for runtime model adaptation.
>
> We consider the cases of applying PTQ and applying QAT for constructing a highly optimized model as the reviewer has mentioned.
>
> - PTQ: While SoTA PTQ methods demonstrate competitive performance with the full precision model up to 4 bits, further reduction in precision leads to significant performance degradation. For example, GuidedQuant [7], a SoTA non-uniform quantization method, shows a large perplexity increase of 1.45 when quantizing Llama-3-8B to 3 bits compared to the full precision model. Thus, maintaining a statically optimized model that meets the tightest margin (i.e., low average precision) would incur a large loss in accuracy when a larger inference slack is available.
>
> - QAT: QAT methods require a tremendous amount of GPU time, which could be impractical considering the rapid advancements of recent LLMs. Since model evolutions are accelerating, the needs for low-cost and fast-deployable quantization methods are rising.
>
> [7] Kim et al., “GuidedQuant: Large Language Model Quantization via Exploiting End Loss Guidance”, ICML, 2025.
>
>
> ---
>
> > **#6**. Do LLM inference slack really have such fluctuations?
>
> In short, yes. As explained in Section 2.2, LLM inference slack is determined by the difference between the QoS budget and system utilization, both of which dynamically vary. We provide additional references to support such variance in QoS budget and system utilization.
>
> - System utilization: Mendis et al. [8] show that memory bandwidth utilization varies both within a workload and across workloads in smartphones. Furthermore, it is common for multiple tasks to be concurrently running on edge devices, including both foreground applications launched by the user and background jobs. Prior work such as ApproxDet [9] specifically takes into account this fluctuation, proposing an adaptive solution that is aware of such resource contention.
>
> - QoS budget: It is well known that different applications of LLMs, while sharing the same model, can have varying performance priorities and requirements [10, 11]. For instance, summarizing a long, complex document may prioritize accuracy over throughput, while live translation requires low time between tokens for better user experience.
>
> [8] Mendis et al., “Impact of Memory Frequency Scaling on User-centric Smartphone Workloads”, SAC, 2018.
>
> [9] Xu et al., “ApproxDet: Content and Contention-Aware Approximate Object Detection for Mobiles”, SenSys, 2020.
>
> [10] Agrawal et al., “On Evaluating Performance of LLM Inference Systems”, arXiv, 2025.
>
> [11] Zhong et al., “DistServe: Disaggregating Prefill and Decoding for Goodput-optimized Large Language Model Serving”, OSDI, 2024.
>
> ---
>
> > **#7**. Other fine-grained neural architecture modifying schemes are not mentioned.
>
>
> While pruning-based methods, including Once-for-All [12] and EagleEye [13] mentioned by the reviewer, can derive multiple sub-networks from a full-sized model for model adaptation, such techniques primarily target non-LLM models, which have several orders of magnitude fewer parameters. Although some pruning methods have been developed specifically for LLMs [14, 15], they generally incur more performance degradation than quantization when applied at an equivalent model compression ratio.
>
> [12] Cai et al., “Once-for-All: Train One Network and Specialize it for Efficient Deployment”, ICLR, 2020.
>
> [13] Li et al., “EagleEye: Fast Sub-net Evaluation for Efficient Neural Network Pruning”, ECCV, 2020.
>
> [14] Ma et al., “LLM-Pruner: On the Structural Pruning of Large Language Models”, NeurIPS, 2023.
>
> [15] Ashkboos et al., “SliceGPT: Compress Large Language Models by Deleting Rows and Columns”, ICLR, 2024.
>
> ---
>
> > **#8**. The precision of activations is ambiguous.
>
> As we employ a weight-only quantization scheme, the activations are kept at high precision (i.e., FP16).
>
> ---
>
> > **#9**. Is the error estimation accuracy a theoretical statement?
>
> Limiting the relative error estimation difference within 8% with 95% confidence is a theoretical statement. From Equation (3), to get the vector norm, we take the square root of every term. To match $\sqrt{(1+\epsilon)}=1.08$, we set $\epsilon=0.1664$. For 95% confidence, we set $\delta=0.05$, and from $k=\mathcal O(\epsilon^{-2}log(\delta^{-1}))$, we can see that $k=64$ is a reasonable selection to match the confidence and error estimation difference statement.
>
> ---
>
> > **#10**. The use of geomean in section 6.3 is concerning.
>
> In Section 6.3 (Line 335-336), we state that “Even at the 99th percentile, the geomean increase remains below 3%.” The geometric mean is calculated from the three values in the 99th percentile row of Table 6 (i.e., 3.02%, 2.25%, and 3.32%), which is 2.83%. We did not intend to disguise any outliers, but used the geometric mean to provide a condensed overview of Table 6.

---

> > ### Comment · Reviewer_WtEe · 2025-08-03
> >
> > Thank you for your response to the points raised in my review. Regarding #3, the perplexity improvement seems rather marginal between the activation norm and the relative error norm. Do you have any insights into why that is?
> >
> > Regarding #8, considering weight-only quantization implies that compute is performed at high precision (e.g., FP16). The concern is then that the paper’s analysis focuses on dynamic weight precision averaged between 3 and 5 bits to claim inference speedups, but reducing activations from 16 to just 8 bits will almost certainly yield higher inference speedups, even in the decode stage. There are several works that show W4A8 can be done for models like those in the paper.
> >
> > References [10] and [11] from your response do support the claim that inference slack is known to present opportunities, which addresses a concern; however, [10] analyzes this across LLM applications (note that each could have different models) and [11] analyzes this across the prefill vs. decode stage. Neither reference supports the claim that dynamic bit width allocation improves inference speed in the decoding stage. First, as in lines 286-287, the paper focuses primarily on the decoding stage and prefills at the highest precision, so the inference slack identified in [11] is not relevant to the argument. Furthermore, it is not clear that dynamic bit allocation addresses the inference slack problems presented in [10]. One example from [10] is that code completion applications might prioritize time-to-last-token (TTLT) whereas conversational agents need consistent time-between-tokens (TBT). In such an example, dynamically allocating the bit width of LLM weights may not sufficiently address the difference between TTLT and TBT optimization objectives. Finally, DP-LLM is not analyzed across LLM applications (which would make the case more convincing), but over a single model for one application.
> >
> > Regarding #9, I’m confused how this is a precise and rigorous theoretical statement (vs a back of the envelope calculation) if you are using the big O notation to make the calculation? Are you taking the precise values of the constants into account or just ignoring them? In that case how can one even compute confidence intervals?

---

> > > ### Author Response · Authors · 2025-08-05
> > >
> > > Thank you for your prompt and insightful response.
> > >
> > > ---
> > >
> > > > **Regarding #3**. Insights for performance resemblance between activation norm and relative error norm.
> > >
> > > As discussed in Section 5.1 (Lines 218–222), about half of the layers exhibit strong correlations between activation norm and relative error norm, which explains why an estimator based solely on activation norm performs reasonably well. Note that we were actually aware of this optimization opportunity and proposed a simpler error estimator that applies linear regression to the activation norm for such layers, which we refer to as the hybrid method.
> > >
> > > Nonetheless, approximately half of the layers still require relative error norm-based estimation for improved performance. While the benefits of this more sophisticated estimator may not be dramatic, they are consistent and clear, so there is little reason not to adopt it, especially given that it introduces no significant runtime overhead.
> > >
> > > ---
> > >
> > > > **Regarding #6**. It is unclear whether DP-LLM is indeed relevant to the scenarios provided by [10] and [11].
> > >
> > > **Inference slack variances in prefill stages**:  It is true that DP-LLM might not be relevant to optimizing the prefill stage. Since DP-LLM is built on a weight-only quantization scheme, runtime model adaptation via DP-LLM does not impact prefill latency.
> > >
> > > Meanwhile, we would like to mention that DistServe [11] also emphasizes time-per-output-token (TPOT) SLO (i.e., decode time) as a critical concern, specifically noting that TPOT requirements may differ even for the same-sized models, based on their target workload. This is precisely the type of runtime adaptation challenge that DP-LLM is designed to address.
> > >
> > > **Metric-specific scenarios**: Runtime model adaptation via DP-LLM can help optimize various performance metrics, such as TTLT and TBT. Although TTLT includes both prefill and decode time, in many cases, decode time dominates due to the autoregressive nature of LLM inference. This implies that runtime model adaptation via DP-LLM, which primarily targets the decode phase, can be particularly effective. Additionally, such adaptation can help achieve more consistent TBT. Without runtime model adaptation, TBT may fluctuate significantly with varying system utilization, whereas runtime model adaptation using DP-LLM allows dynamic bitwidth adjustment in response to these fluctuations, leading to more uniform performance.
> > >
> > > **Supporting multiple LLM applications**: For deploying multiple applications, using a shared backbone model combined with lightweight LoRA adapters is a practical and efficient approach for on-device LLM deployment, enabling fast task adaptation while minimizing storage overhead. In this scenario, DP-LLM can be applied to the backbone to tailor effective bitwidth to each application’s requirements. This allows diverse applications to be deployed efficiently through adapter-specific specialization atop a unified, quantized backbone.
> > >
> > > Even for the cases where separate models are deployed per application, DP-LLM remains relevant, as QoS budgets can still vary within a single application. For instance, a chatbot may require low latency for casual conversation but can tolerate higher latency for summarization or complex queries. In these situations, runtime adaptation via DP-LLM still can play an important role.
> > >
> > > ---
> > >
> > > > **Regarding #8.** Reducing activation bits from 16 to 8 will yield inference speedups.
> > >
> > > Reduction in activation bits does not lead to meaningful decoding speedups for low- and single-batched scenarios. As DP-LLM targets on-device LLM inference, where single-batch inputs from a single user are dominant, linear layers are memory bandwidth-bounded. In such scenarios, loading weights from memory is the primary bottleneck [4, 16]; thus, the change in activation bits results in little to no performance gain.
> > >
> > > This is well demonstrated in the roofline analysis of QServe [16]. W4A16 and W4A8 share the same memory rooflines under small batch sizes, thus having the same performance.
> > >
> > > [16] Lin et al., “QServe: W4A8KV4 Quantization and System Co-design for Efficient LLM Serving”, MLSys, 2025.
> > >
> > > ---
> > >
> > > > **Regarding #9**. The use of big-O notation for calculation is not a rigorous statement.
> > >
> > > We acknowledge the non-precise use of big-O notations, and we empirically analyze the C4 dataset. We find that the error estimation difference can be done within a 15% difference with 91% confidence.
> > >
> > > We thank the reviewer for enhancing the rigor of our paper and will rectify the mistake in the revised version.

---

> > > > ### Comment · Reviewer_WtEe · 2025-08-05
> > > >
> > > > Thank you for your responses. I will carefully consider them as I decide on my final rating.

---

### Official Review · Reviewer_mMxs · 2025-07-03

**Clarity:** 3
**Significance:** 2
**Originality:** 3
**Rating:** 4
**Confidence:** 3

**Summary:**

This paper identifies a new perspective in LLM quantization: while previous works consider layer-wise sensitivity as static, the authors observe that the sensitivity of each layer varies across decoding iterations. Motivated by this, they propose DP-LLM, a runtime quantization strategy that dynamically selects bitwidths per layer at each decoding step, taking an instance-specific view of quantization. The method includes a principled mechanism to allocate bit budgets per layer under an average precision constraint, along with lightweight error estimators to ensure quality guarantees. Extensive experiments demonstrate that DP-LLM achieves improved performance compared to static mixed-precision methods under the same average bitwidth, validating the benefit of dynamic, layer-wise quantization.

**Questions:**

1.	DP-LLM performs dynamic, per-layer, per-token bitwidth selection during autoregressive decoding. This implies that, for any given forward pass, a layer may require either W_l or W_h.
Could the authors clarify how weights are stored and accessed at runtime? Specifically:Is DP-LLM implemented using a Matryoshka quantization approach, where only the highest-precision weights (e.g., 6-bit) are loaded, and lower-precision weights are derived on-the-fly via truncation?
2.	Other concerns：
For additional questions and clarifications regarding runtime assumptions, implementation details, and ablation completeness, please refer to the identified weaknesses above.

**Ethical Concerns:**

["NO or VERY MINOR ethics concerns only"]

**Final Justification:**

Some of my concerns have been addressed by the authors' response. I think conducting more extensive experiments on latency overhead would strengthen the work. Additionally, I recommend providing a more detailed discussion or empirical verification on ideas to extend DP-LLM to multi-batch settings.

**Limitations:**

Yes

**Quality:**

3

**Strengths And Weaknesses:**

Strengths:

1. The paper is generally well-organized and well-written.

2. The core insight—that layer-wise quantization sensitivity varies across decoding iterations—is original and not addressed in prior work.

3. The newly proposed dynamic per-layer precision allocation effectively improves model performance while satisfying the same average bitwidth constraint.

4. The authors conduct extensive experiments on two recent large models (LLaMA-3-8B and Phi-3-Medium) across multiple datasets and downstream tasks.

Weakness:

1. While the idea of per-iteration, per-layer precision assignment is theoretically appealing, the current design of DP-LLM makes it difficult to support standard batched inference across multiple queries. Although autoregressive generation is inherently token-by-token within each query, it is common in practice to process multiple queries in parallel by batching them at each decoding step. However, DP-LLM selects bitwidths dynamically for each layer and each decoding iteration based on the input vector of that specific query. This leads to query-dependent computation paths and weight selection, which prevents efficient batching due to divergent bitwidth configurations across queries. In contrast, static mixed-precision approaches assign fixed bitwidths per layer and are thus fully compatible with batch-level parallelism, making them more practical for high-throughput deployment.

2. Although the proposed dynamic precision selection aims to improve the accuracy-latency trade-off, the paper does not provide a thorough comparison of latency overheads against strong static baselines such as LLM-MQ and HAWQ-V2. Given that DP-LLM involves runtime computations for precision selection—either via relative error estimation or linear predictors—it is important to quantify how much latency these mechanisms add relative to static mixed-precision methods, which do not incur such runtime costs. A more detailed latency analysis would strengthen the empirical justification of DP-LLM’s design and better inform practical deployment decisions.

3. The formulation of the integer programming problem in Appendix A—particularly Equation (6)—lacks important clarification regarding Q. Especially, the relationship between Q and b is unclear. If Q is assumed to be fixed in advance, it remains unclear how the decision variables c_{i,b} can be optimized.

4. In Algorithm 1, the computation of the output y is expressed as a weighted sum over bitwidth candidates, but the current form effectively reduces toy = 1 * W_{i,b}, since only one c_{i,b}  is active (equal to 1). This appears inconsistent with the linear interpolation form described in the main text (e.g., Line 194), where y is computed as a convex combination of W_{i,b}x and W_{i,b+1} x,. It is unclear whether this discrepancy is an intentional simplification, a design decision only for the integer programming phase, or simply a typo. Clarifying this would help improve the rigor and consistency of the method presentation.

5. The paper includes a per-query QoS analysis showing that DP-LLM introduces small deviations between actual and target bitwidths across queries (as in Table 6). However, it remains unclear whether such deviations are unique to DP-LLM or if similar fluctuations would occur in static methods like LLM-MQ or HAWQ-V2.

---

> ### Author Rebuttal · Authors · 2025-07-31
>
> Thank you for your insightful comments.
>
> ---
>
> > **#1**. How does DP-LLM handle multi-batch scenarios?
>
> DP-LLM is specifically designed for single-batch scenarios, which we believe to be the dominant use case in on-device LLM inference, where a single query comes from a single user [1]. Nevertheless, DP-LLM can be extended to support multi-batch scenarios, for example by averaging the relative error across batched inputs. However, this may smooth out the dynamic sensitivity of individual queries, potentially diminishing performance gains. We believe that developing extensions of DP-LLM for multi-batch settings in data centers, while minimizing such degradation, represents a promising direction for future work.
>
> [1] Lin et al., “AWQ: Activation-aware Weight Quantization for On-Device LLM Compression and Acceleration”, MLSys, 2024.
>
> ---
>
> > **#2**. Latency overheads are not discussed.
>
> Table 4 reported in the paper shows the latency overhead against LLM-MQ. Latency overheads are measured while error estimation mechanisms are concurrently running. The overhead against HAWQ-V2 is omitted since static baselines have virtually the same latency under equal effective bitwidth. Additionally, Table 5 shows the effect on latency when utilizing different relative error estimation techniques.
>
> ---
>
> > **#3**. Clarification for Q in Equation (6) of Appendix A is lacking.
>
> $W_Q$ should be $W_{i,b}$ in Equation (6). The underlying quantizer, which is Any-Precision LLM for the current implementation, will create every $W_{i,b}$ in advance. Given such quantized weights, the goal of Equation (6) is to decide $c_{i,b}$ to minimize the loss perturbation.
> We apologize for the ambiguity.
>
> ---
>
> > **#4**. Algorithm 1 is incoherent with the main text.
>
> It is simply a typo. $s_{i,b}W_{i,b}x+t_{i,b}W_{i,b}x$ is supposed to be $s_{i,b}W_{i,b}x+t_{i,b}W_{i,b+1}x$. We apologize for the confusion.
>
> ---
>
> > **#5**. Is the per-query QoS deviation unique to DP-LLM?
>
> Per-query QoS deviation is unique to DP-LLM. While static methods have equal average bitwidths over all decoding steps, DP-LLM aims to match the average bitwidth among multiple decoding steps to the target precision. This may result in small deviations in the actual average bitwidth used against the target precision.
>
> ---
>
> > **#6**. Is DP-LLM implemented using a Matryoshka quantization approach?
>
> Yes, for evaluations, we build DP-LLM on top of Any-Precision LLM, which adopts the Matryoshka quantization approach. This eliminates the need to store separate models for each bitwidth; instead, it maintains only the high-bitwidth parameters, from which lower-bitwidth variants are derived via on-the-fly truncation.

---

> > ### Comment · Reviewer_mMxs · 2025-08-03
> >
> > Thank you for the authors' response. Conducting more extensive experiments on latency overhead would strengthen the work. Additionally, I recommend providing a more detailed discussion or empirical verification on ideas to extend DP-LLM to multi-batch settings. As my original rating is already positive, I have decided to keep it unchanged.

---

### Official Review · Reviewer_QAUB · 2025-07-03

**Clarity:** 3
**Significance:** 3
**Originality:** 3
**Rating:** 4
**Confidence:** 4

**Summary:**

This paper introduces DP-LLM, a dynamic layer-wise precision assignment method for quantized LLMs. The key idea is to dynamically adjust each layer’s quantization bitwidth during decoding iterations based on estimated relative error, instead of relying on static precision assignment. The results show improvements in perplexity and downstream tasks over different baselines, with minimal latency overhead.

**Questions:**

1.Does this method work with batch sizes larger than one? If so, how do you handle situations where tokens in the same batch might need different bitwidths?
2.While the trade-off between performance and memory is intuitive, how is the trade-off between performance and latency manifested in practice?
3.The framework has multiple components—thresholds, error estimators, asynchronous steps, etc. How much does each part contribute to the final results? Are all of them really needed, or could the system be simplified?

**Ethical Concerns:**

["NO or VERY MINOR ethics concerns only"]

**Final Justification:**

The response provide important additional information for this work (e.g., the trade-off between performance and latency manifested). I decide to raise my score.

**Limitations:**

Yes.

**Paper Formatting Concerns:**

None.

**Quality:**

2

**Strengths And Weaknesses:**

**Strength**

1.The idea of adjusting bitwidths dynamically for each token is quite innovative and makes sense for balancing performance and efficiency.
2.The paper is well structured, with a clear explanation of the method and easy to understand.
3.The writing is generally clear, and the explanations are easy to follow。
4.The authors provide code and helps others reproduce and build on this work.

**Weaknesses**

1.The analysis in section 2.4 lacks detailed quantification. Figure 3 shows dynamic sensitivity, but it doesn’t explain which types of tokens are sensitive to which layers, or which layers are most affected. It’s also unclear which dataset was used for this figure and whether the results generalize to other tasks and models.
2.The experiments are mainly done in a PyTorch setup, but real-world systems often use frameworks like vLLM. It’s unclear how the method would perform there, especially for latency . Also, the latency ablation study does not provide detailed performance changes for different setups.
3.The method seems quite complex, but the performance improvements shown in the tables are modest. Also, there’s no information about variability or statistical significance, so it’s hard to judge how reliable the gains are. This raises questions about whether the method is practical to deploy.

---

> ### Author Rebuttal · Authors · 2025-07-31
>
> Thank you for your thoughtful review.
>
> ---
>
> > **#1**. Detailed quantification of Section 2.4 is needed. Also, clarification for Figure 3 is needed.
>
> As noted in Line 127–128 of the paper, Figure 3 was generated using the first sample from the C4 train dataset, decoded with the Llama-3-8B model.
>
> To provide quantitative insights into the dynamic nature of sensitivity, we present statistics based on the experimental data in Figure 3 that captures how dynamically the set of sensitive layers changes across decoding steps. Specifically, we compute the overlap of sensitive layers (i.e., the top 20% most sensitive layers) across all pairs of decoding steps. The average overlap ratio is only 4.83%. For reference, if the sensitive layers were selected completely at random, the expected overlap would be 4% (i.e., 0.2 × 0.2). The closeness between the two values (4.83% vs. 4%) suggests that the set of sensitive layers varies almost as much as a purely random selection, highlighting the importance of addressing this dynamism to achieve better performance.
> To further analyze the dynamics of sensitive layer distribution, we present two additional analyses based on the same experimental data in Figure 3.
>
> The first addresses the question: _“Is there a consistent trend where the same tokens lead to similar sets of sensitive layers?”_ To investigate this, we compute the average overlap ratio of sensitive layers for repeated occurrences of the same token. We select the ten most frequent tokens in the sample for analysis (',', '.', 'to', 'will', 'BBQ', 'in', 'you', 'be', 'and', 'the'), and present the results in the following table. The maximum average overlap ratio observed is only 6.32%, which is not significantly different from the random baseline of 4% (i.e., 0.2 $\times$ 0.2). This indicates that even for identical tokens, the set of sensitive layers varies significantly depending on the context. Therefore, the answer to the question is inclined toward _No_. This motivates the need to account for such dynamic behavior, thereby justifying the design and value of our approach.
>
> | Token | , | . | to | will | BBQ | in | you | be | and | the |
> |:---:|:---:|:---:|:---:|:---:|:---:|:---:|:---:|:---:|:---:|:---:|
> | Ratio | 4.99% | 4.96% | 4.91% | 4.78% | 5.65% | 4.24% | 4.32% | 5.28% | 6.32% | 5.06% |
>
> The second explores the question: _“Do different layers tend to exhibit different levels of sensitivity?”_ Specifically, we examine sensitivity variations across layer types (`q_proj`, `k_proj`, `v_proj`, `o_proj`, `gate_proj`, `up_proj`, and `down_proj`). We compute the **selection ratio**—the proportion of decoding steps in which a given layer is selected as sensitive—for each layer, and then average these ratios by layer type. The results show that `down_proj` has the highest average selection ratio at 27.71%, while `k_proj` demonstrates the lowest at 16.97%. Notably, the `down_proj` of the final transformer block (block 31) achieves the highest selection ratio across all layers, reaching 45.27%. These findings reveal a discernible trend in sensitivity differences across layer types.
>
> Such observations suggest that static mixed-precision methods may still offer some degree of effectiveness, as sensitivity shows partial correlation with layer type and can be partially anticipated. However, this effectiveness is limited at best—even the most frequently selected sensitive layer is chosen in only 45.27% of decoding steps. This means that statically fixing this layer as sensitive would result in selecting a suboptimal layer in more than half of the decoding steps, missing opportunities to allocate precision where it is most needed. These findings once again underscore the importance of dynamic approaches like DP-LLM.
>
> ---
>
>
> > **#2**. Real-world frameworks such as vLLM are not used for evaluation.
>
> We believe the current latency evaluation setup is appropriate for measuring the overheads of DP-LLM. For this evaluation, we used **gpt-fast**, a highly optimized LLM inference implementation specifically designed for small batch sizes. This makes it well-suited for DP-LLM, which primarily targets single-batch on-device inference. In contrast, frameworks such as **vLLM** are optimized for datacenter serving scenarios with large batch inputs, and are therefore less relevant to the intended use case of DP-LLM.
>
> ---
>
> > **#3**. Statistical significance is unclear.
>
> Two sources of variability for DP-LLM may exist: 1) the initialization point of each average precision when the fine-tuning process starts, and 2) the randomly sampled $A$ matrix when applying JL-Lemma. Once a configuration is ready for evaluation, all computations do not involve any randomness and result in a deterministic outcome.
>
> The following table presents the average and standard deviation of DP-LLM perplexity using Llama-3-8B, evaluated across five different random seeds. The results are compared to those reported in the original paper. Only minimal variations are observed across runs.
>
> |  | Target Precision | Reported Perplexity | Average Perplexity | Standard Deviation |
> |:---:|:---:|:---:|:---:|:---:|
> | WikiText2 | 3.5 | 7.00 | 7.01 | 0.00948 |
> |  | 4.0 | 6.59 | 6.59 | 0.01007 |
> |  | 4.5 | 6.41 | 6.43 | 0.01016 |
> | C4 | 3.5 | 11.03 | 11.05 | 0.01511 |
> |  | 4.0 | 10.25 | 10.25 | 0.00972 |
> |  | 4.5 | 9.97 | 9.98 | 0.01575 |
>
> ---
>
> > **#4**. How does DP-LLM handle multi-batch scenarios?
>
> DP-LLM is specifically designed for single-batch scenarios, which we believe to be the dominant use case in on-device LLM inference, where a single query comes from a single user [1]. Nevertheless, DP-LLM can be extended to support multi-batch scenarios, for example, by averaging the relative error across batched inputs. However, this may smooth out the dynamic sensitivity of individual queries, potentially diminishing performance gains. We believe that developing extensions of DP-LLM for multi-batch settings in data centers, while minimizing such degradation, represents a promising direction for future work.
>
> [1] Lin et al., “AWQ: Activation-aware Weight Quantization for On-Device LLM Compression and Acceleration”, MLSys, 2024.
>
> ---
>
> > **#5**. How is the trade-off between performance and latency manifested?
>
> Latency is directly proportional to memory consumption and thus exhibits the same trade-off with performance. In general, slower models tend to yield better performance. The following table illustrates this trade-off by showing the time-per-output token (TPOT) for different effective bitwidths.
>
>
> | Device | Effective Bitwidth | 3.5 | 4.0 | 4.5 | FP16 |
> |:---:|:---:|:---:|:---:|:---:|:---:|
> | RTX 4060Ti | Llama-3-8B | 16.26ms | 17.94ms | 19.62ms | 55.43ms |
> |  | Phi-3-Medium | 24.83ms | 28.18ms | 31.28ms | OOM |
> | Jetson | Llama-3-8B | 30.18ms | 33.49ms | 36.23ms | 86.36ms |
> |  | Phi-3-Medium | 45.64ms | 51.28ms | 58.21ms | 158.73ms |
>
> ---
>
> > **#6**. How much does each part contribute to the final results?
>
> Every component of our framework is integral to the performance and efficiency of the system.
>
> The ablation study of the error estimator is available in Table 3. The ablation study for using asynchronous error estimation is in Table 5. An ablation study regarding thresholds is not possible, since dynamic adaptation per decoding step would become infeasible without them.

---

> > ### Comment · Reviewer_QAUB · 2025-08-05
> >
> > Thanks for the detailed response, which have addressed my major concern.

---

### Note · Authors · 2025-08-13

We sincerely thank all reviewers and ACs for their time, effort, and valuable feedback. While we appreciate the constructive discussions, we would like to highlight four key clarifications.

---

> **1. LLM Inference slacks do fluctuate.**

As discussed in our response to #6 of reviewer WtEe, prior works show that on-device inference often experiences varying QoS budgets and system utilization, leading to fluctuating inference slack. Efficiently adapting to these variations is a crucial challenge for on-device LLMs. DP-LLM addresses this with a practical and innovative approach that exploits the overlooked opportunity of _dynamic sensitivity_.

---

> **2. Fine-grained static mixed precision is unsuitable for runtime adaptation.**

We compared DP-LLM to two static layer-wise mixed precision methods (LLM-MQ and HAWQ-V2). While more fine-grained approaches (e.g., channel-wise, block-wise) exist, their runtime overhead makes them impractical for real-time adaptation. As noted in our responses to #4 of reviewer WtEe and #3 of reviewer sm1G, SliM-LLM—a block-wise mixed precision method—incurs significant latency overhead. Meeting tight latency requirements is critical for runtime adaptation, rendering these static fine-grained methods incompatible.

---

> **3. DP-LLM meets latency requirements.**

DP-LLM is built on _gpt-fast_, a framework optimized for low-batch, on-device inference. As shown in our responses to #5 of reviewer QAUB and #6 of reviewer sm1G, DP-LLM achieves latency reductions proportional to bitwidth decreases, enabled by our low-overhead error estimation techniques. This ensures both realistic implementation and practicality for deployment.

---

> **4. DP-LLM leverages unexplored opportunities.**


As noted by reviewers, our use of dynamic sensitivity to enable token-wise mixed precision is novel. Unlike prior static saliency-based methods with fixed precision assignments, DP-LLM adapts precision dynamically across decoding steps. Our response to #1 of reviewer QAUB provides a more quantitative and extensive analysis of this dynamic behavior.

---

We thank the reviewers and ACs again and will incorporate the constructive feedback into our revised version.

---

### Decision · Program_Chairs · 2025-09-17

**Decision:**

Accept (poster)

**Comment:**

(a) Summary of Claims

The paper introduces DP-LLM, a runtime mechanism for dynamic, token-wise, layer-wise precision assignment in large language models. Unlike static mixed-precision methods, DP-LLM adapts bitwidths per layer during decoding based on lightweight error estimation, aiming to optimize the latency–accuracy trade-off under fluctuating runtime constraints.

(b) Strengths

- Novelty: First to exploit dynamic sensitivity across decoding steps for runtime precision adaptation.
- Practicality: Lightweight design compatible with on-device inference; minimal overhead.
- Empirical depth: Covers multiple models (8B, 70B), low-bit scenarios, latency analysis, and ablations.
- Reproducibility: Code availability and detailed methodology.

(c) Weaknesses

- Theoretical rigor: Original JL-based confidence bound was informal; authors moved to empirical calibration.
- Scope limitation: Focused on single-batch, on-device inference; batching scenarios remain unexplored.
- Latency reporting: Needs consolidated presentation and clearer comparisons.

(d) Reasons for Decision

The paper proposes a clear and original idea with strong practical relevance and demonstrates convincing empirical results across diverse settings. While some formal and presentation issues remain, they are addressable in the camera-ready version. The rebuttal significantly strengthened the paper with additional experiments and clarifications. Overall, the contribution is technically sound, well-motivated, and impactful for efficient LLM deployment.

(e) Discussion & Rebuttal

Reviewer concerns:
– Lack of detailed analysis of dynamic sensitivity and token-level behavior.
– Latency overhead and fairness of comparisons.
– Scalability to large models and low-bit regimes.
– Clarity of equations and theoretical justification for error estimation.
Author responses:
– Provided quantitative analysis of sensitivity dynamics.
– Added latency tables for RTX 4060Ti and Jetson; overhead shown to be minimal.
– Extended experiments to LLaMA-3-70B and low-bit (2–4b) scenarios; included storage-fair comparisons vs OWQ.
– Clarified algorithmic typos and reframed JL bound empirically.